# Brain songs framework used for discovering the relevant timescale of the human brain

Gustavo Deco[1,2,3,4], Josephine Cruzat [1,5] & Morten L. Kringelbach[5,6]

A key unresolved problem in neuroscience is to determine the relevant timescale for understanding spatiotemporal dynamics across the whole brain. While resting state fMRI reveals networks at an ultraslow timescale (below 0.1 Hz), other neuroimaging modalities such as MEG and EEG suggest that much faster timescales may be equally or more relevant for discovering spatiotemporal structure. Here, we introduce a novel way to generate whole-brain neural dynamical activity at the millisecond scale from fMRI signals. This method allows us to study the different timescales through binning the output of the model. These time-scales can then be investigated using a method (poetically named brain songs) to extract the spacetime motifs at a given timescale. Using independent measures of entropy and hierarchy to characterize the richness of the dynamical repertoire, we show that both methods find a similar optimum at a timescale of around 200 ms in resting state and in task data.

[1] Center for Brain and Cognition, Computational Neuroscience Group, Department of Information and Communication Technologies, Universitat Pompeu Fabra, Roc Boronat 138, Barcelona 08018, Spain. [2] Institució Catalana de la Recerca i Estudis Avançats (ICREA), Passeig Lluís Companys 23, Barcelona 08010, Spain. [3] Department of Neuropsychology, Max Planck Institute for Human Cognitive and Brain Sciences, 04103 Leipzig, Germany. [4] School of Psychological Sciences, Monash University, Melbourne, Clayton, VIC 3800, Australia. [5] Department of Psychiatry, University of Oxford, Oxford OX3 7JX, UK. [6] Center for Music in the Brain, Department of Clinical Medicine, Aarhus University, Aarhus DK-8000, Denmark. Correspondence and requests for materials should be addressed to G.D. (email: gustavo.deco@upf.edu) or to M.L.K. (email: morten.kringelbach@psych.ox.ac.uk)

I t is well-known from physics that the relevant timescale of a complex system depends in a non-linear way on the timescale of the individual components[1]. In brain science, a prototypical example is the demonstration that the timescale of a randomly connected population of neurons is not the same as that of the individual neurons[2]. At the whole-brain level, the problem of the relevant time-scale is significantly non-trivial given the heterogeneous timescales of the many types of neural elements and synapses. Furthermore, the neural elements are not randomly connected across the whole-brain but are shaped by a very particular underlying anatomical connectivity[3]. Overall, this creates a difficult and yet unsolved question of finding the relevant time-scale for discovering the spatiotemporal structures underlying whole-brain processing. A further complication is the current lack of appropriate measurement equipment for obtaining different spatial and temporal information across the whole human brain. For example, traditional neurophysiology only allows access to multiple neurons at the milliseconds level but not at the whole-brain level. In contrast, neuroimaging can provide whole-brain information but is restricted in the temporal domain to measuring haemodynamic activity on the scale of seconds (functional magnetic resonance imaging, fMRI) or somewhat limited in the spatial domain (encephalography, EEG or magnetoencephalography, MEG)[4].

The question of relevancy of a given timescale is dependent on the experimental measurements and the question of interest. As an example, think of the difference between weather and climate, where some experimental data such as rain is measured on the timescale of minutes and hours, while wet seasons are measured on the timescale of months, and climate change on the timescale of decades[5]. Thus, if we are interested in the prediction of rainfall, the relevant timescale of experimental data is over minutes and hours, while measurements taken over, say, seconds, months or years are not particularly helpful.

The key question investigated here is finding the relevant timescale for obtaining the spatiotemporal structures underlying whole-brain dynamics to reflect the temporal evolution of spatial brain networks. More specifically we use independent component analysis (ICA) to estimate the spatial co-activation patterns and track the activity of these patterns over time. This description of the temporal evolution of underlying spatial networks we here call spacetime motifs. In particular, we study the relevant timescale for maximising the richness of repertoire of spacetime motifs, i.e. by studying the entropy of the switching activity between all possible motifs in a brain state (see Methods). In order to be able to answer this question, we require quantitative measurements of whole-brain activity across timescales from milliseconds to seconds and minutes.

Over the last decade, much progress has been made in identifying brain processing of information by assuming that a good quantitative account of brain processing of information can be found in spatial brain maps obtained from fMRI data[6,7]. In particular, resting state networks (RSN) on the ultraslow (below 0.1 Hz) timescale have been identified using a variety of techniques under the resting state condition, i.e. without stimulation or task[8]. Nevertheless, it is not clear that this spatial information of RSNs provides the full description of the rich repertoire of spacetime motifs across the human brain. Other researchers have started to use MEG, which provides much faster information on the milliseconds timescale and have found similar spatial RSNs to those found at the ultraslow timescale[9–11]. Yet, the similarity in terms of the static spatial information may not be the most relevant. Interestingly, a Hidden Markov Model (HMM) has been used to identify faster states in MEG with some spatial similarities to RSNs but importantly found states with a lifetime of around 200 ms[12]. Still, the authors did not investigate to what extent

these fast brain states are involved in creating a more or less rich repertoire of spacetime motifs.

Furthermore, the question of timescale has been explored using neuronal avalanches[13,14]. The results suggest that the representation of specific objects is not likely to be found at the temporal scale of avalanches, but rather at the scale of their sequences[14], perhaps similar to those found for spacetime motifs.

In order to continue the advance in resolving the question of timescale in humans and at the whole-brain level, we developed a method, tentatively named 'brain songs' with poetic license as an extension of the previous historically named 'cortical songs' and 'cell assemblies'[15–17] but perhaps better called 'extraction of whole-brain spacetime motifs'. This methodology combines two concepts: (1) using whole-brain computational modelling of neuroimaging timeseries to recover the underlying neurodynamical timeseries of the data (in milliseconds) and (2) estimating the significant spacetime motifs emerging at a given timescale and to use the whole-brain measures of entropy and hierarchy to estimate the relevance and richness of the underlying dynamical repertoire. Here, we use an extension of our previous ignition-based hierarchical measures[18,19] (see Methods). The spacetime motifs are the whole-brain spatiotemporal patterns resulting from this process. Using this novel application for extracting spacetime motifs at the whole-brain level, we can study broadcasting of information across the whole brain over all timescales using measures of entropy, hierarchy and dynamic functional connectivity (FC), going beyond existing methods merely estimating static spatial maps. As such it is using state-of-the-art statistical methods to allow a systematic investigation of many timescales from milliseconds to seconds[20]. This method can provide evidence for the optimal timescale underlying the human brain's rich spatiotemporal dynamical repertoire. More generally, our novel approach allows empirical investigations to recover the many timescales of neural signals at the whole-brain level, thus significantly improving on the natural constraints of the tools being used currently.

We describe the discovery of a fundamental finding for theoretical and experimental neuroscience, namely an optimum timescale of around 200 ms using whole-brain modelling of resting state and task data and characterizing the richness of the dynamical repertoire using independent measures of entropy and hierarchy. The causal mechanistic results reveal that the relevant timescale of the human brain emerges from network properties and specifically the structural connectivity coupling. Furthermore, the proposed novel brain songs framework could help resolve the underlying spatiotemporal dynamics of brain processing for other brain states.

## Results

**Discovering the timescale of human brain processing**. We were interested in determining the optimal timescale for discovering relevant spatiotemporal structures maximising the richness of repertoire of spacetime motifs underlying human brain processing. We used whole-brain modelling to generate whole-brain neural activity at any timescale by fitting the model to existing neuroimaging BOLD fMRI data and generating timescales from milliseconds to seconds by binning the output of whole-brain model's underlying millisecond timescale.

We combined and extended existing methods for extraction of spatiotemporal structures (spacetime motifs) from this whole-brain neural activity, allowing us to quantify the richness of the repertoire of the human brain.

The full pipeline is shown in Fig. 1, demonstrating how whole-brain computational modelling constrained by the underlying anatomical connectivity (diffusion MRI, dMRI) can fit the BOLD

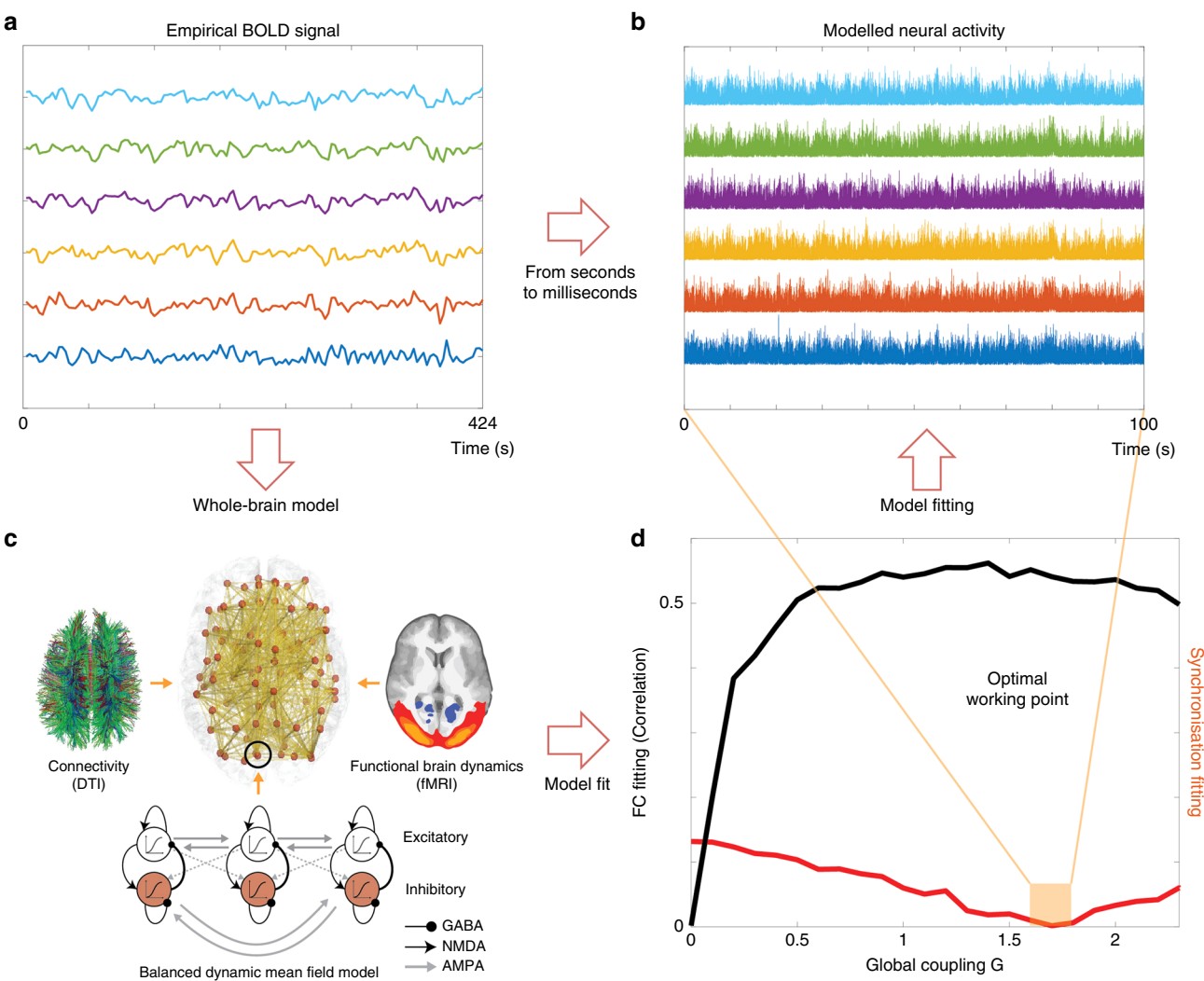

**Fig. 1** General whole-brain modelling scheme for generating milliseconds time series from BOLD data. **a** Extracting the BOLD time series from fMRI with a typical time scale of 2 s. **b** These BOLD time series are generated by neural activity on the scale of milliseconds. This scale is fundamental to be able to investigate the relevant time scale of brain activity. **c** The transformation from slow to fast time scale is accomplished by whole-brain modelling, where we use the slow data to fit a balanced dynamic mean field model with realistic synaptic dynamics and shaped by the underlying anatomical skeleton. **d** The optimal working point of the model is found using the optimal global synchronisation level (red line shows the quadratic error of the difference between the empirical and simulated Kuramoto order parameters, see Methods) and fitting to the static functional connectivity (FC, black line shows the correlation between the empirical and simulated static FC matrices). At the optimal working point (corresponding to the minimum of the synchronisation fit, shown in orange box), the model generates the milliseconds time series which is used to find the relevant time scale

fMRI timeseries (Fig. 1a). Since the underlying model is a realistic neuronal model including AMPA, GABA and NMDA receptors (see Methods and Fig. 1c, d), we are able to generate the neuronal timeseries at the milliseconds level for the optimal working point of the model (Fig. 1b) fitted to the empirical BOLD fMRI data (Fig. 1d) using the FC and, most importantly, the global synchronisation level (see Methods); precisely constraining the working point of the model. We wanted to show the robustness of this methodology and therefore on purpose chose to use a dataset with relatively few participants (16), long TR of 3.03 s and relatively short resting state duration (~7 min).

Figure 2 shows the precise algorithm used for the extraction of whole-brain spacetime motifs obtained from the binned data at different timescales. The first step is summarised in Fig. 2a which is to binarise the averaged time bin neuronal data by extracting events following the established procedures of Tagliazucchi[21] and for the ignition method[18,19]. In short, an event for a given brain

region is defined by binarising the transformed averaged time bin neuronal time series into z-scores $z_i(t)$ and imposing a threshold $\theta$ such that the binary sequence $\sigma_i(t) = 1$ if $z_i(t) > \theta$, and is crossing the threshold from below, and $\sigma_i(t) = 0$ otherwise. Please note that this method is threshold-independent as shown by Tagliazucchi[21] given that it uses the so-called Poincaré section (see Methods).

In Fig. 2b, c we show how spacetime motifs are extracted using the established method for detecting neuronal assemblies from spike data[20]. Briefly summarising this method, which uses three main steps: (1) Construction of the event matrix, where events are binned; (2) Determination of the number of spacetime motifs, as the eigenvalues above the maximum of the eigenvalues of the null hypothesis distribution based on random matrix theory, following the Marčenko–Pastur distribution[22]; and (3) Extraction of spacetime motifs using ICA and estimation of corresponding activity, where co-activation patterns are found and used to track

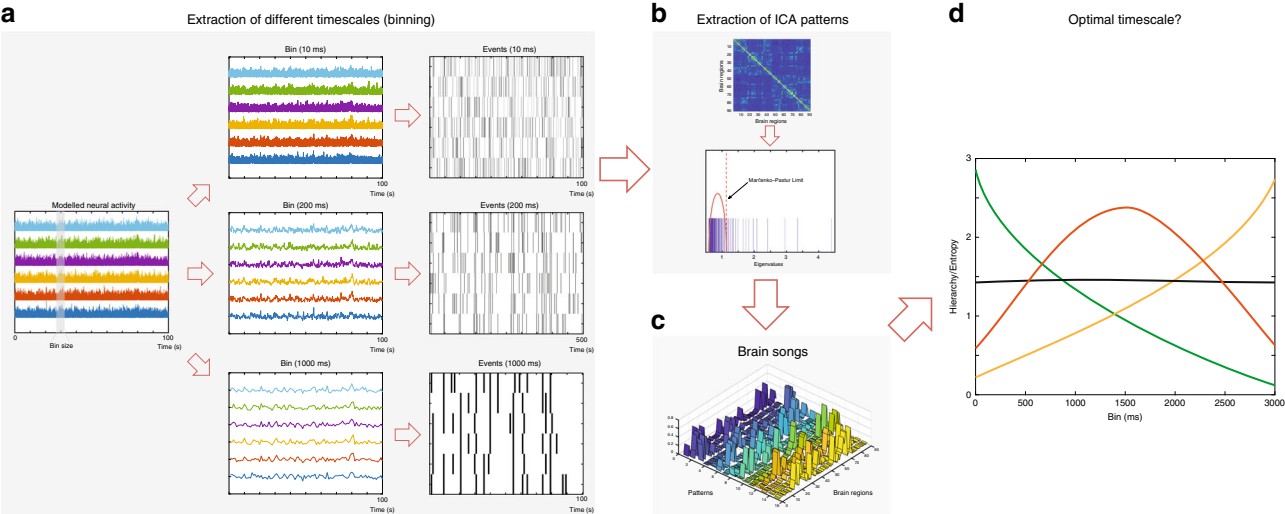

**Fig. 2** Extraction of spacetime motifs. **a** In order to study the relevant time scale, we create different bin sizes of the milliseconds neural time series. The middle panel shows the data with 10, 200, 1000 ms bin sizes. These time binned time series are binarised using a point-process algorithm (shown on the right). **b** In order to extract the number of significant spacetime motifs, we compute the eigenvalues above the maximum of the eigenvalues of the null hypothesis distribution based on random matrix theory, following the Marčenko–Pastur distribution[22]. **c** We then extract the spacetime motifs using independent component analysis (ICA) and estimate the corresponding activity, where co-activation patterns are found and used to track the activity over time. **d** The richness of the dynamical repertoire at different timescales can be computed from the spacetime motifs and corresponding probabilities allow using measures of entropy and hierarchy of functional brain organisation (see Methods). We show the four possible different scenarios of how this may vary with timescale whether flat, monotonic decrease or increase or having an optimum

the activity over time of the spacetime motifs. In other words, the spacetime motifs are the non-thresholded components extracted by ICA.

The spacetime motifs and corresponding probabilities allow us to compute the richness of the dynamical repertoire at different timescales as measured with entropy and hierarchy of functional brain organisation (see Methods). There are essentially four different possible scenarios of how this may vary with timescale, whether flat, monotonic decrease or increase or having an optimum (Fig. 2d).

The entropy characterises explicitly the richness of the repertoire from a probabilistic perspective. Complementarily, the measure of hierarchy is a variation on the ignition-based hierarchical measures[18,19]. The main idea is to define the relevancy of each brain region for the broadcasting of information across the whole brain. For this, we define the 'cohesiveness' of each brain region as the summation of the product of three things for a given spacetime motif: the degree of participation, the probability and the 'broadness' defined as its size (see Methods). The hierarchy is the degree of diversity of brain regions according to the level of cohesiveness, or more formally the standard deviation of the cohesiveness across all brain areas.

Figure 3a shows the entropy and hierarchy measured as a function of the timescale (binning size). We show the spacetime motifs for four timescales [very fast (10 ms), optimal (200 ms), slow (1000 ms) and very slow (2000 ms)] in terms of the Transition Probability Matrix (TPM, Fig. 3b), probability state space (Fig. 3c) and the patterns of the extracted spacetime motifs (Fig. 3d).

As can be seen from Fig. 3a, we found that both the entropy (red line) and hierarchy (blue line) measures consistently achieve a maximum at around 200 ms, while the static grand average FC (orange line) monotonically increases. This confirms our assumption that grand average FC measures are not well suited for characterising the relevant timescale for brain processing, whereas entropy and hierarchy are clearly highly appropriate

measures for discovering the relevancy of having a rich dynamical repertoire at a given timescale.

For the relevant spacetime motifs emerging at 200 ms, it is remarkable how uniformly distributed the individual states are in terms of their probability of occurrence (see Fig. 3c). In contrast, in the other timescales of 10, 1000 and 2000 ms, the distribution of individual states is much less evenly distributed, suggesting that certain states are dominating and thus leading to impoverished dynamical repertoires. Furthermore, at these suboptimal timescales there are fewer spacetime motifs (11, 10 and 9 states) than at optimal timescale of 200 ms (15 states). This suggests that we could be missing crucial information if we try to characterise the spatiotemporal networks at the wrong timescale.

The individual spacetime motifs at 200 ms are shown rendered on the standard brain in Fig. 4. As can be seen, these brain states resemble known RSNs, e.g. networks 6 and 15, which correspond to the frontal part of the default mode network and the visual network, respectively. The other networks resemble sub-components and lateralised versions of the classical RSNs. We stress here the resemblance rather than any formal equivalence to RSNs.

Given that the timescale of 2000 ms must be closely related to the timescale used for classical resting state analyses of BOLD data, we show in Fig. 5 a comparison of the visual network and frontal part of the default mode network between the brain states found in the spacetime motifs at 2000 ms and at 200 ms. As can be seen, the networks are very similar but of course the underlying state probability is different. Thus the previous methodology used for extracting RSNs are valid for finding the spatial components but of course less suitable for extracting the underlying temporal dynamics given the inherent temporal constraints of BOLD signals. The results presented here offer a potential solution to extracting both time and space structure from the richness of the repertoire involved in brain processing.

After having shown that methodology clearly works with a relatively challenging dataset, we then set about to confirm the

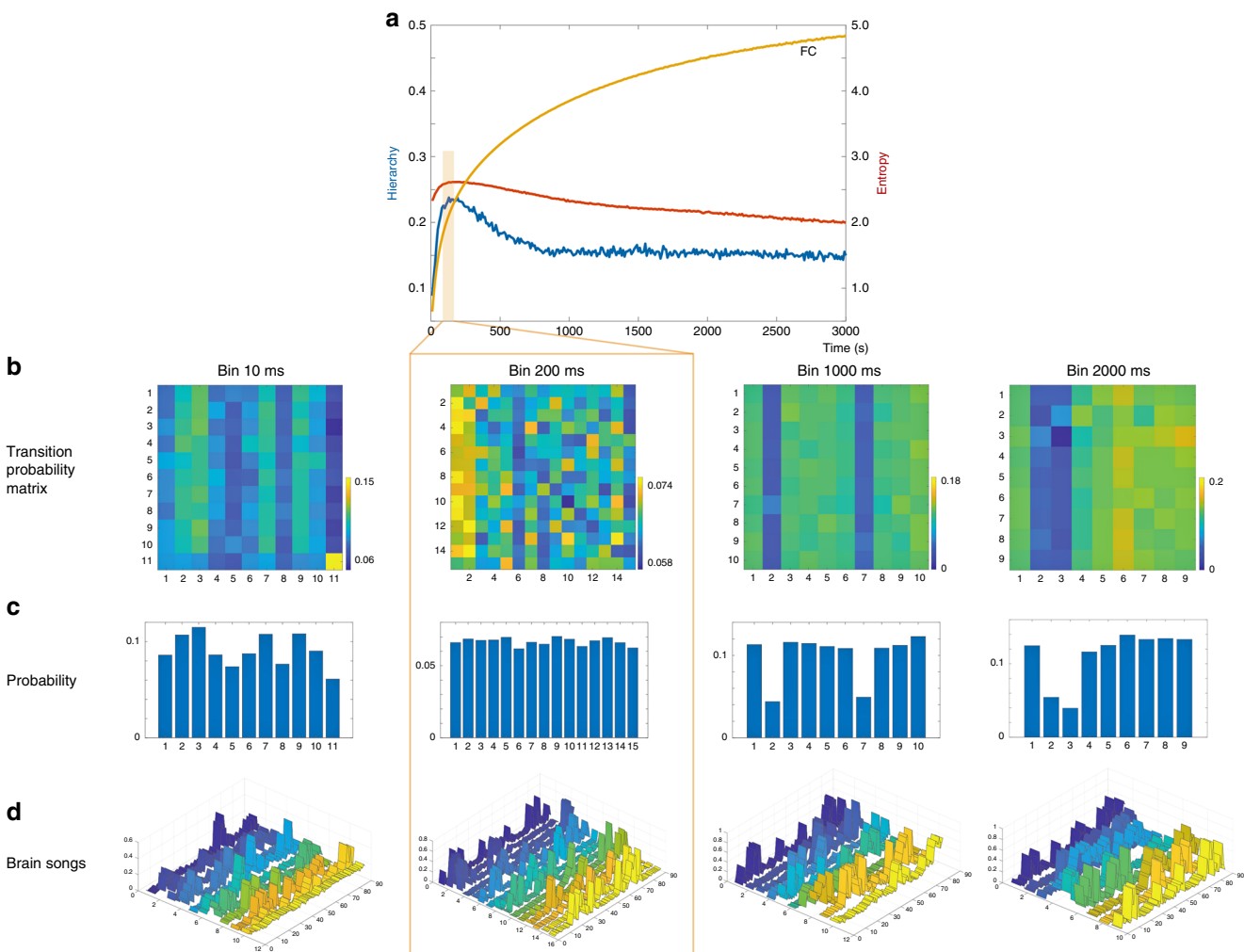

**Fig. 3** Discovering the relevant timescale of the brain. **a** Using the methods outlined in the two other figures, we here show the results of using these on normal human resting state activity. In particular, we show the results of using three different measures (entropy, red line; hierarchy, blue line; and mean functional connectivity, FC, orange line) on the data in different bin sizes from 10 to 3000 ms. As can be seen very clearly from the peaks in entropy and hierarchy (red and blue lines), the richness of the dynamical repertoire is found in the region of around 200 ms (light orange box). Please note that the mean FC is monotonically increasing, suggesting that this static measure is not ideal for finding the relevant time scale of the dynamic rich repertoire of brain states. We show the spacetime motifs for four timescales [very fast (10 ms), optimal (200 ms), slow (1000 ms) and very slow (2000 ms)] in terms of the **b** Transition Probability Matrix, **c** probability state space and **d** the patterns. At the timescale of 200 ms, it is remarkable how uniformly distributed the individual states are in terms of their probability of occurrence, which is in contrast to the other timescales of 10, 1000 and 2000 ms, where there are also fewer spacetime motifs

reliability by using state-of-the-art datasets. Supplementary Figure 1 shows the reliability of the results by analysing the data from the freely available high-reliability, high-quality Human Connectome Project (HCP) data of 100 unrelated subjects with 15 min resting state and a much faster TR of 0.72 s (see Methods). In addition to confirming the timescale for resting state data, we were also interested in discovering whether timescale is modified under task conditions. We therefore analysed the social cognition task data from the same 100 unrelated HCP subjects. In the social cognition task, participants were presented with short video clips (20 s) of objects (squares, circles, triangles) that either interacted in some way, or moved randomly on the screen. Figure 6 shows the entropy and hierarchy measured as a function of the timescale (binning size). The results show a similar peak around 200 ms to that found in resting state data (compare with Fig. 3a and Supplementary Figure 1). This confirms that timescale found in resting state is also found in task condition, suggesting that the timescale is an intrinsic property of whole-brain dynamics.

Bolstering the findings of the 200 ms timescale found in resting state and task, we analysed resting state neuroimaging MEG data which has the right timescale and thus does not require whole-brain modelling step. Figure 7 shows spacetime motifs in MEG data and in particular the entropy and hierarchy as a function of the timescale (binning size). We split the MEG data into different delta, theta, alpha and beta bands. As can be seen, both measures peak at 200 ms for all bands, strongly confirming our finding with whole-brain modelling of fMRI BOLD signals.

In order to ascertain the robustness of the results, we have carried out many additional analyses. First, in Supplementary Figure 2, we found the same consistent maximum at around 200 ms when running the full analysis using three different threshold for the binarisation process. This clearly demonstrates the robustness of the results and the independence of the threshold binarisation method as expected given the use of the threshold-independent Poincaré section.

Second, we further investigated the dependence of the results on the parameters of the whole-brain model. Generally, the

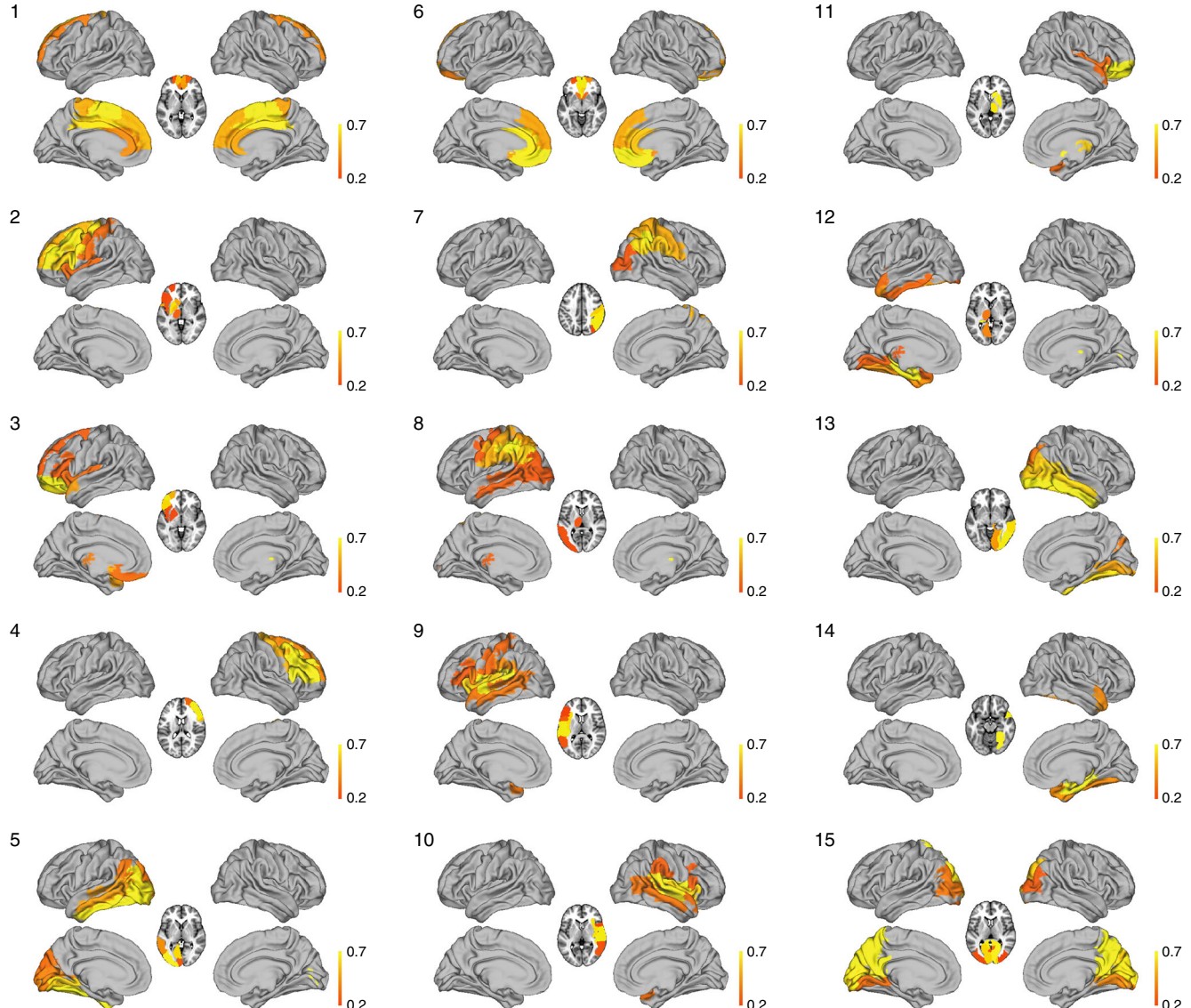

**Fig. 4** Individual spacetime motifs at 200 ms rendered on the standard brain. Some of these brain networks resemble known resting state networks, e.g. networks 6 and 15, which correspond to the frontal part of the default mode network and the visual network, respectively. The other networks resemble sub-components and lateralised versions of the classical resting state networks, namely: 1. medial cingulate, 2. left orbitofrontal, 3. left prefrontal, 4. right prefrontal, 5. left higher order visual areas, 6. frontal DMN, 7. right parietal, 8. left parietal, 9. left auditory and insula, 10. right STG, auditory and insula, 11. right orbitofrontal, 12. left hippocampus, 13. right higher order visual areas, 14. right hippocampus, 15. visual network

results must depend on two main factors: local node dynamics and whole-brain coupling. In terms of the local node dynamics, we use biophysically realistic parameters (see Methods). We changed two main parameters: (A) the ratio of excitation to inhibition at the local regional level and (B) the biophysical latencies of the NMDA of the local dynamics to a non-biological value. As can be seen in left and right panels of Supplementary Figure 3, this breaks the timescale by removing the peak. More importantly, we carried out extensive simulations of the whole-brain coupling in terms of shifting the working point of the whole-brain model (shown in Supplementary Figure 4) and damaging highest edge couplings in the whole-brain network (shown in Supplementary Figure 5). As can be seen from the plot of hierarchy in Supplementary Figure 4 and Supplementary Figure 5, the timescale optimum is not found, when these brain network parameters are different from the empirical data. In summary, only the realistic whole-brain model consistent with

the empirical data is showing the relevant timescale discovered in this paper. This suggests that the relevant timescale emerges from the structural connectivity coupling of the human brain.

## Discussion

In this paper, we have investigated a central question in neuroscience, namely which is the most relevant timescale for brain processing, i.e. broadcasting and making information available across the whole-brain, i.e. information as the intrinsic transmission of activity—and not the neural encoding of information about sensory inputs, behaviour or cognition. We first fit a whole-brain dynamic mean field (DMF) model with realistic pools of excitatory and inhibitory neurons as well as synaptic dynamics to BOLD fMRI data and generate the underlying timeseries on the timescale of milliseconds. Secondly, we bin this data on different timescales and for each of these measures, we compute the

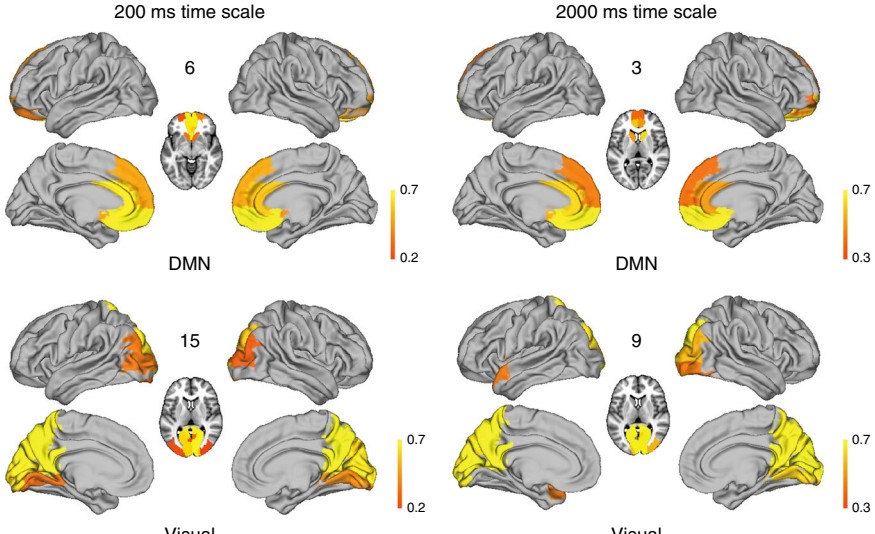

**Fig. 5** Comparison of resting state networks at optimal (left, 200 ms) and slow BOLD (right, 2000 ms) timescales. The figure shows the similarity of the spatial characteristics of the spacetime motifs networks found at both timescales. As an example, the visual network (bottom) and frontal part of the default mode network (top) are shown. Thus, classical methodologies used for extracting resting state networks are valid for finding the spatial components but of course less suitable for extracting the underlying temporal dynamics given the inherent temporal constraints of BOLD signals

independent measures of entropy and hierarchy to characterise the dynamical repertoire. In order to demonstrate the usefulness of the methodology for the general neuroimaging community, we deliberately chose a dataset with relatively few participants, long TR and relatively short resting state duration.

The results were very clear with a convergence for both measures on an optimum at the timescale of around 200 ms (Fig. 3), and we also replicated these findings in the much larger, state-of-the-art HCP dataset with 100 unrelated participants, very short TR and long resting state duration (Supplementary Figure 1). Importantly, when testing other timescales, the measures show that these are non-optimal. In particular we extracted spacetime motifs for three examples of non-optimal timescales [very fast (10 ms), slow (1000 ms) and very slow (2000 ms)] which—unlike the spacetime motifs found at the optimal 200 ms timescale—are all unbalanced in probability space of occurrence, suggesting an impoverishment of the dynamical repertoire.

Instead, the spacetime motifs at the optimal 200 ms timescale bear close resemblance to classical RSNs (Fig. 4) but, similarly to networks with fast time dynamics extracted from MEG[12], they are split into subcomponents. Other MEG studies have tried to find the relevant timescale[9–11] but assumed that only grand average spatial correlations were important and thus were able to replicate the fMRI RSNs. We speculate that the spacetime motifs found at 200 ms may be the building blocks of brain function.

While not the main aim here, we also analysed resting state neuroimaging MEG data and confirmed an optimal 200 ms timescale for delta, theta, alpha and beta bands (Fig. 7). Crucially, this analysis used only the second step of our new methodology, namely extracting whole-brain spacetime motifs, given that the excellent time resolution of MEG means that we do not have to use the first step involving constructing a whole-brain model but crucially validates the novel first step of our procedure using whole-brain modelling for accessing much faster timescales using slow fMRI BOLD data.

So, as we show here, the relevant timescale is not in the order of tens of seconds but rather faster at 200 ms which is consistent with previous findings of microstates in EEG[23,24] and when using HMM on MEG data[12]. Importantly, building on microstate and

HMM findings from EEG and MEG, here we are able to determine that these fast spatiotemporal dynamics are in fact optimal in creating the richest spatiotemporal dynamics. Further investigations combining and comparing our procedure in fMRI and MEG could shed new light on the role of timescale in different brain states.

Furthermore, the results also provide further mechanistic insights into the question of timescale given that our findings are based on a causal mechanistic whole-brain model. When the results change after directly manipulating elements of the mechanistic model, this directly establishes a causal relationship. We manipulate every part of the model to establish a causal relationship between a given element and the findings. In particular, we manipulated the three main elements to the whole-brain model: (1) the local regional dynamics (Supplementary Figure 3), (2) the global working point of the dynamics (the only free parameter, G, see Methods and Supplementary Figure 4) and (3) the underlying anatomical skeleton (Supplementary Figure 5). All of these manipulations yielded a change in radical breakdown of the timescale optimum, thus establishing a causal relationship between the underlying mechanisms and the intrinsic whole-brain timescale of 200 ms.

In the context of thinking about the relevance for conscious brain state, the main finding of a relevant timescale of 200 ms for brain processing in both resting state and task data is very interesting. The research of Dehaene, Changeux and colleagues have consistently shown that there is a critical bottleneck for conscious processing with information typically being processed and broadcast for conscious processing at this 200–250 ms timescale[25–27]. For example, Dehaene and colleagues have shown using a simple backward masking task with a flashed stimulus that broadcasting related to conscious reportability is related to ignition of activity at around 270 ms in a highly distributed fronto-parieto-temporal networks, while the subliminal trials did not elicit such ignition[25]. Even stronger data from neural multi-recordings in non-human primates show that ignition of visual stimuli were associated with strong sustained activity in dorso-lateral prefrontal cortex around 200–250 ms when reported, while this frontal activity was weaker and quickly decayed for unreported stimuli[27]. These findings are typically interpreted in terms

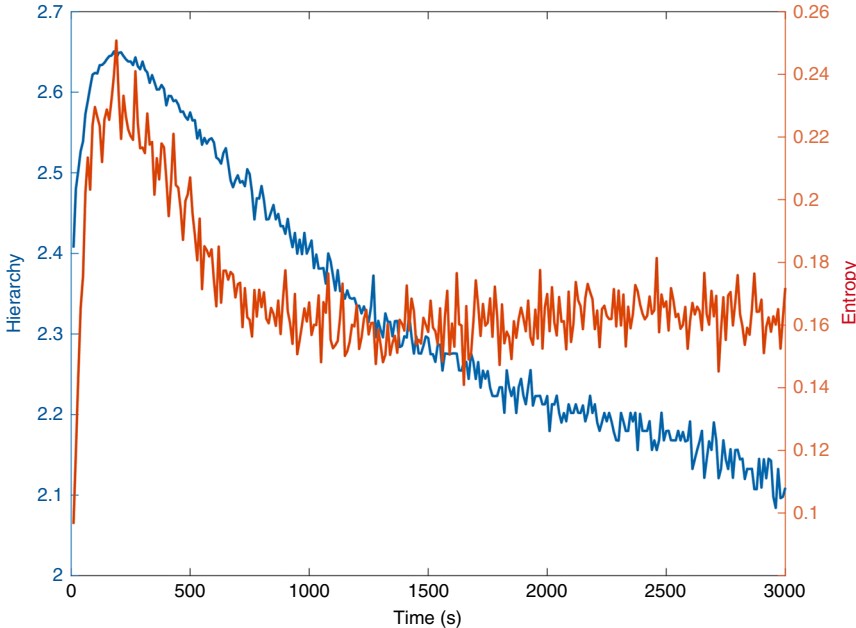

**Fig. 6** Timescale in HCP task data. We analysed HCP task data (see Methods) in order to compare the optimal timescale for resting state and neuroimaging during a social cognition task, where participants were presented with short video clips (20 s) of objects (squares, circles, triangles) that either interacted in some way, or moved randomly on the screen. We fitted the whole-brain model to the BOLD signal and plot the results of measuring entropy (red line) and hierarchy (blue line) in different bin sizes from 10 to 3000 ms. We found peaks in entropy and hierarchy of around 200 ms, which is very similar to the peak found in resting state fMRI. This means that the timescale for resting state and task condition is optimal at the same timepoint, suggesting that the 200 ms timescale is an intrinsic property of brain dynamics

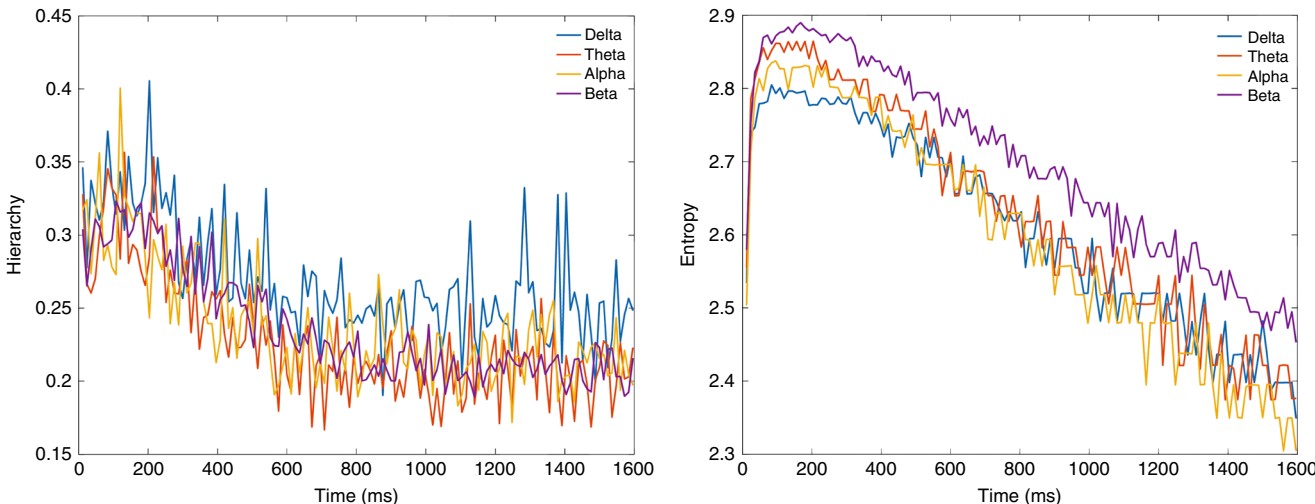

**Fig. 7** Timescale in MEG data. Figure shows the results of using hierarchy (left) and entropy (right) on the different delta (1–4 Hz), theta (4–8 Hz), alpha (8–12 Hz) and beta (12–30 Hz) bands of the MEG data in different bin sizes from 10 to 1600 ms, i.e. directly on the empirical data—and thus not using the whole-brain modelling part of the process used for fMRI data. The peaks in entropy and hierarchy for all bands show a peak in the region of around 200 ms, which is similar to the peak found in fMRI and thus suggest a similar richness of the dynamical repertoire

of the global workspace theory[28,29], and this experimental support, together with our present findings, suggest that the timescale for this global workspace must be around 200 ms. Importantly, this has to be sustained by a set of regions binding and broadcasting information, as has been shown in work investigating the functional relevance of individual brain regions within whole-brain networks[30].

Similarly, the results are also consistent with other theories of consciousness such as the Integrated Information Theory (IIT)[31] and the Temporo-spatial Theory of Consciousness (TTC)[32]. The 200 ms timescale could be fundamental for enhancing the effects

of integration of information necessary for consciousness. Speculatively, the timescale of 200 ms may provide the fundamental harmonic frequency for the temporo-spatial nestedness serving as the neural predisposition of consciousness mentioned in the TTC theory[32].

The fact that the timescale finding is conserved across experimental conditions suggests that it might be a general principle of brain function. Yet, the exact relationship between timescale and brain function remains to be clarified, and future research could investigate in timescale in altered brain states such as sleep or anaesthesia.

The optimal timescale of 200 ms raises the interesting question of the underlying neural oscillations with theta (1-Hz) might play a significant role. Unfortunately, the whole-brain DMF model used here is based on the fMRI BOLD signal which does not contain fast oscillations in ranges above 0.05 Hz, which is obviously much slower than theta and related rhythms. In addition, the DMF model is asynchronous, yet still the 200 ms timescale emerges which would argue against theta and related faster rhythms playing a significant role.

The present timescale findings are complementary to the important body of work on neuronal avalanches[13,33,34]. Perhaps, as has been suggested, the timescale of avalanches is not to be found at the temporal scale of the avalanches, but rather at the scale of their sequences[14]. As such the relevant timescale described here and the spacetime motifs could be related to this scale of sequences of neuronal avalanches. Even more, these two phenomena—criticality of neuronal avalanches and optimal timescale—coexist at the subcritical working point of the whole-brain model[35,36]. Interestingly, the ITT and TTC theories mentioned above are also supported by evidence of scale-freeness and neuronal variability in different conscious states such as awake[37,38] and anaesthesia[39,40].

Furthermore, it has been shown that the properties of the duration and size of avalanches change at different levels of brain state[41]. Since brain activity can have different levels of synchronisation over time[42], perhaps the optimal temporal scale also depends on the underlying brain state with low synchronisation (REM sleep, moments of high attention), whereas in periods of high synchronisation (slow wave sleep, drowsiness) the optimal temporal scale is slightly different, perhaps due to the higher incidence of silences in unit activity[43].

Our new paradigm opens new lines of exciting research. Focusing just on the measures of entropy and hierarchy, it would be of considerable interest to use these on data with milliseconds resolution, i.e. EEG, MEG and iEEG data, to use these measures and the whole-brain model to establish causal links to microstates and HMM and to replicate the main finding here of a relevant timescale of 200 ms. As shown in Fig. 7, our preliminary analyses of MEG data found a similar timescale of 200 ms for this faster neuroimaging methodology but much in depth research remains to be done.

Regarding the novel whole-brain modelling method for generating the underlying millisecond timeseries from BOLD data, it would also be of tremendous interest to investigate potential changes in timescale (position of optimum) and richness of repertoire (entropy and hierarchy at optimal timescale) in task and in different brain states (e.g. sleep, anaesthesia, coma and minimal consciousness) and neuropsychiatric diseases. Our preliminary results, shown in Fig. 6, show that task data has a similar optimal timescale but much remains to be explored. In general, we hypothesise that the timescale and richness of repertoire are unbalanced in brain disorders, albeit in different ways for different disorders. Importantly, given that such investigations would be using a whole-brain model, this could subsequently be used to perturb and rebalance the model to find an optimal, causal path to health[44,45].

Overall, the results suggest that the relevant timescale of the human brain emerges from the network properties and specifically the structural connectivity coupling. Long-term, the proposed new method could help resolve the underlying spatiotemporal dynamics of brain processing for other pertinent questions regarding the timescales of fast and slow cognition[46,47].

## Methods

**Overview of methodology**. The extraction of whole-brain spacetime motifs methodology described here combines two important advances: (A) extraction of whole-brain timeseries on multiple timescales (from milliseconds to seconds) from a whole-brain optimised mean-field model of empirical BOLD data (on the timescale of seconds) and (B) extraction of meaningful spatiotemporal dynamics structure (spacetime motifs) by a method generalising previous cell-assembly methods used in neurophysiology. In the following, we describe the specific technical details of this process.

**Neuroimaging data acquisition, preprocessing and timeseries extraction**. Ethics: The neuroimaging part of the study was approved by the internal research board at CFIN, Aarhus University, Denmark and given ethics approval by the Research Ethics Committee of the Central Denmark Region (De Videnskabsetiske Komitéer for Region Midtjylland). All participants gave written informed consent prior to participation.

The Washington University–University of Minnesota (WU-Minn HCP) Consortium obtained full informed consent from all participants, and research procedures and ethical guidelines were followed in accordance with Washington University institutional review board approval.

Participants: We used data from two populations. One group of 16 participants from Aarhus, Denmark and one group of 100 unrelated participants from the publicly available database from the HCP from the WU-Minn HCP Consortium.

The online recruitment system at Aarhus University helped to recruit all 16 healthy right-handed participants (11 men and 5 women, mean age: 24.75 ± 2.54). We screened participants and excluded those with psychiatric or neurological disorders (or a history thereof) from participation in this study. All 16 participants were scanned with MRI and MEG as specified below.

The data set used for this investigation was selected from the March 2017 public data release from the HCP where we chose the sample of 100 unrelated participants (54 females, 46 males, mean age = 29.1 ± 3.7 years). This subset of participants provided by HCP ensures that they are not family relatives. This criterion was important to exclude possible identifiability confounds and the need for family-structure co-variables in the analyses.

Neuroimaging acquisition for MRI (Aarhus): The 3T Siemens Skyra scanner at CFIN, Aarhus University, Denmark was used to collect MRI data (structural MRI, rs-fMRI and diffusion MRI) in one session. The parameters for the structural MRI T1 scan used a voxel size of 1 mm³; reconstructed matrix size 256 × 256; echo time (TE) of 3.8 ms and repetition time (TR) of 2300 ms. The resting-state fMRI data were collected using whole-brain echo planar images (EPI) with TR = 3030 ms, TE = 27 ms, flip angle = 90°, reconstructed matrix size = 96 × 96, voxel size 2 × 2 mm with slice thickness of 2.6 mm and a bandwidth of 1795 Hz/Px. We collected approximately 7 min of resting state data per subject.

For the estimating the structural connectivity we collected dMRI data using TR = 9000 ms, TE = 84 ms, flip angle = 90°, reconstructed matrix size of 106 × 106, voxel size of 1.98 × 1.98 mm with slice thickness of 2 mm and a bandwidth of 1745 Hz/Px. Furthermore, the data were collected with 62 optimal nonlinear diffusion gradient directions at $b = 1500$ s/mm². Approximately one non-diffusion weighted image ($b = 0$) per 10 diffusion weighted images was acquired. The dMRI images were collected with two different phase encoding directions, the first acquisition used anterior to posterior phase encoding direction, while the second acquisition was performed in the opposite direction.

Neuroimaging acquisition for fMRI (HCP): The 100 HCP participants were scanned on a 3-T connectome-Skyra scanner (Siemens). We used one resting state fMRI acquisition of approximately 15 min acquired on the same day, with eyes open with relaxed fixation on a projected bright cross-hair on a dark background. The HCP website (http://www.humanconnectome.org/) provides the full details of participants, the acquisition and preprocessing of the data.

Neuroimaging acquisition for MEG (Aarhus): In addition to the rsMRI and dMRI, the same 16 participants from Aarhus also had resting state MEG (rs-MEG) data acquired using a 306 channel Elekta Neuromag TRIUX system (Elekta Neuromag, Helsinki, Finland) located in a magnetically shielded room at the CFIN at Aarhus University Hospital, Denmark. All data were recorded at a sampling rate of 1000 Hz with an analogue filtering of 0.1–330 Hz. Approximately 5 min of resting state was collected for each participant.

Before data collection, a three-dimensional digitizer (Polhemus Fastrak, Colchester, VT, USA) was used to record the participant's head shape relative to the position of four headcoils, with respect to three anatomical landmarks, which could be registered on the MRI scan (the nasion, and the left and right preauricular points). The structural MRI scan for each participant was acquired during a separate session (see above). The position of the headcoils was tracked during the entire recording using continuous head position identification (cHPI), providing information on the exact head position within the MEG scanner. This allows for accurate movement correction at a later stage during data analysis.

**Neuroimaging structural connectivity and functional timeseries**. Parcellation: All neuroimaging data was processed using the Automated Anatomical Labeling (AAL) parcellation[48], which is perhaps the most widely used parcellation scheme. The AAL90 subset of the 116 regions consists of 76 cortical regions and 14 subcortical regions including the thalamus, basal ganglia, amygdala and hippocampus but leaves out 26 cerebellar regions. A full description and labels of the regions can be found at http://neuro.imm.dtu.dk/wiki/Automated_Anatomical_Labeling.

For the Aarhus dataset, we used the linear registration tool from the FSL toolbox (www.fmrib.ox.ac.uk/fsl, FMRIB, Oxford)[49] to coregister the EPI image to the T1-weighted structural image. The T1-weighted image was co-registered to the

T1 template of ICBM152 in MNI space[50]. The resulting transformations were concatenated and inversed and further applied to warp the AAL template[48] from MNI space to the EPI native space, where interpolation using nearest-neighbor method ensured that the discrete labelling values were preserved. Thus, the brain parcellations were conducted in each individual's native space. Similarly, we mapped the AAL to the cortical surface using the Conte69 template using the Connectome Workbench software[51] and for the subcortical AAL regions we determined the membership of each grayordinate to a given AAL region.

Structural connectivity from dMRI: For estimating the structural connectivity we used FSL diffusion toolbox (Fdt) for the Aarhus diffusion MRI data. We used the default parameters of this imaging pre-processing pipeline on all participants. The local probability distribution of fibre direction was estimated at each voxel. We used the probtrackx tool in Fdt to provide automatic estimation of crossing fibres within each voxel. This has been shown to significantly improve the tracking sensitivity of non-dominant fibre populations in the human brain[52].

The connectivity probability from a seed voxel $i$ to another voxel $j$ was defined by the proportion of fibres passing through voxel $i$ that reach voxel $j$ using a sampling of 5000 streamlines per voxel[52]. This was extended from the voxel level to the region level, i.e. in an AAL90 parcel consisting of $n$ voxels, $5000 \times n$ fibres were sampled. The connectivity probability $P_{ij}$ from region $i$ to region $j$ is calculated as the number of sampled fibres in region $i$ that connect the two regions divided by $5000 \times n$, where $n$ is the number of voxels in region $i$.

We computed the undirected connectivity probability between the 90 regions within the AAL90 parcellation[48] as the average of probabilities of connectivity $P_{ij}$ and $P_{ji}$. For both phase encoding directions, we computed the $90 \times 90$ symmetric weighted network based on the AAL90 parcellation, and normalised by the number of voxels in each AAL region; thus representing the structural connectivity network organization of the brain.

Preprocessing and extraction of functional timeseries in fMRI resting state and task data: For extracting the functional timeseries from Aarhus data, we first preprocessed the resting state fMRI data using the MELODIC toolbox (Multivariate Exploratory Linear Decomposition into Independent Components) Version 3.14[53]. We used the default parameters for the imaging pre-processing pipeline on all participants: motion correction using MCFLIRT[49]; non-brain removal using BET[54]; spatial smoothing using a Gaussian kernel of FWHM 5 mm; grand-mean intensity normalisation of the entire 4D dataset by a single multiplicative factor and high pass temporal filtering (100.0 s). To extract and average the time courses from all voxels within each AAL cluster we used standard FSL tools.

For the HCP resting state and task datasets, the data preprocessed by the HCP using standardized methods using FSL (FMRIB Software Library), FreeSurfer, and the Connectome Workbench software[51,55]. Briefly, it included correction for spatial and gradient distortions and head motion, intensity normalization and bias field removal, registration to the T1 weighted structural image, transformation to the 2 mm Montreal Neurological Institute (MNI) space, and using the FIX artefact removal procedure[55,56]. The head motion parameters were regressed out and structured artefacts were removed by ICA + FIX processing (ICA followed by FMRIB's ICA-based X-noiseifier[57,58]). We then used a custom-made matlab script with the ft_read_cifti function to extract the average timeseries of all the grayordinates in each AAL90 region.

Extraction of MEG data timeseries from AAL regions: For extracting the MEG data timeseries in the AAL90 regions, we downsampled the raw MEG sensor data (204 planar gradiometers and 102 magnetometers) from 1000 to 250 Hz using MaxFilter and converted this data to SPM8 format. For cleaning and removing potential artefacts from the MEG data, we used the methods described in our previous paper[59]. We then used the AAL90 template to parcellate the brain. A scalar implementation of the LCMV beamformer was applied to estimate the source level activity of the MEG sensor data at each brain area[60–62]. We co-registered each participant's structural T1-weighted MRI scan to the standard MNI template brain using affine transformation, and further referenced to the space of the MEG sensors by use of the Polhemus head shape data and the three fiducial points. An overlapping-spheres forward model was computed, representing the MNI-co-registered anatomy as a simplified geometric model using a basis set of spherical harmonic volumes[63]. We directed the beamformer at locations corresponding to center-of-gravity coordinates of the AAL90 regions[64]. The beamformer was applied to the broadband sensor data band-pass filtered between 2 and 40 Hz, where the two sensor modalities (magnetometers and planar gradiometers) were combined by normalising them by the mean of their respective eigenvalues. The estimation of the data covariance matrix, necessary for the beamformer reconstruction, was regularised using the top 62 (minus number of rejected ICs) principal components. This yielded, for each data set, a [90 × number of samples] source-space data matrix, representing the spontaneous activity at the 90 AAL areas in the 2–40 Hz frequency range. We extracted the timeseries in the delta (1–4 Hz), theta (4–8 Hz), alpha (8–12 Hz) and beta (12–40 Hz) bands.

**(A) Extraction of arbitrary timescale using whole-brain modelling.** Below, we describe the full details of the balanced whole-brain mean field model (Fig. 1) and the methods for fitting the model to the empirical BOLD data comparing the fit using standard average FC and the Kuramoto synchronisation index (Fig. 2).

Whole-brain mean field model: The brain dynamics sustained by the underlying empirical dMRI-based anatomical connectivity was based on the whole-brain model of Deco and colleagues[65]. This model describes the functional

dynamics of local regions for a given brain parcellation. The dynamics of each local brain regions is given by excitatory–inhibitory sub-networks (E–I networks), which are mutually interconnected according to the underlying anatomical connections. The anatomical structural connectivity matrix is obtained from diffusion-imaging data from healthy human subjects as described above. Under resting state conditions, each single brain node emulates spontaneous neuronal noise, i.e. reproducing the typical asynchronous low firing rate spontaneous activity observed empirically (around 3 Hz for the pyramidal neurons and 9 Hz for inhibitory neurons,). We implement this neuronal noise by a modified DMF model based on the original reduction of Wong and Wang[66]. In this DMF reduction, the excitatory synaptic currents are mediated by NMDA receptors and the inhibitory currents are mediated by GABA-A receptors. The inhibitory pools are connected reciprocally with the excitatory pools but only locally, whereas the excitatory pools are coupled by long-range connections based on the dMRI Structural Matrix ($C_{ij}$). The structural matrix $C_{ij}$ denotes the density of fibres between brain area $i$ and $j$. More specifically, the DMF model of the whole brain can be expressed by the following system of coupled differential equations:

$$I_i^{(E)} = W_E I_0 + w_+ J_{NMDA} S_i^{(E)} + GJ_{NMDA} \sum_j C_{ij} S_j^{(E)} - J_i S_i^{(I)}, \quad (1)$$

$$I_i^{(I)} = W_I I_0 + J_{NMDA} S_i^{(E)} - S_i^{(I)}, \quad (2)$$

$$r_i^{(E)} = H^{(E)}\left(I_i^{(E)}\right) = \frac{a_E I_i^{(E)} - b_E}{1 - \exp\left(-d_E\left(a_E I_i^{(E)} - b_E\right)\right)}, \quad (3)$$

$$r_i^{(I)} = H^{(I)}\left(I_i^{(I)}\right) = \frac{a_I I_i^{(I)} - b_I}{1 - \exp\left(-d_I\left(a_I I_i^{(I)} - b_I\right)\right)}, \quad (4)$$

$$\frac{dS_i^{(E)}(t)}{dt} = -\frac{S_i^{(E)}}{\tau_E} + \left(1 - S_i^{(E)}\right)\gamma r_i^{(E)} + \sigma v_i(t), \quad (5)$$

$$\frac{dS_i^{(I)}(t)}{dt} = -\frac{S_i^{(I)}}{\tau_I} + r_i^{(I)} + \sigma v_i(t), \quad (6)$$

Here, $r_i^{(E,I)}$ denotes the population firing rate of the excitatory (E) or inhibitory (I) population in the brain area ($i$). $S_i^{(E,I)}$ denotes an excitatory (E) or inhibitory (I) synaptic gating variable in the local area ($i$). The input currents to the excitatory (E) or inhibitory (I) population ($i$) are given by $I_i^{(E,I)}$. The population firing rates are ($H^{(I)}$ and $H^{(E)}$) of the input synaptic currents to the excitatory or inhibitory population $i$, given by $I^{(I)}$ and $I^{(E)}$. Parameter values for the neuronal response input–output functions $H$ are: $a_E = 310$ (VnC), $b_E = 125$ (Hz) and $d_E = 0.16$(s) for the excitatory pool and $a_I = 615$ (VnC), $b_I = 177$ (Hz) and $d_I = 0.087$(s) for the inhibitory pool. The kinetic parameters are $\gamma = 0.641/1000$ (the factor 1000 expresses rate-constants in ms), and $\tau_E = \tau_{NMDA} = 100$ (ms) and $\tau_I = \tau_{GABA} = 10$ (ms). The overall effective external input is $I_0 = 0.382$ (nA) with $W_E = 1$ and $W_I = 0.7$. Furthermore, $w_+ = 1.4$ is the local excitatory recurrence. These parameters are from Wong and Wang[66], which are in turn derived from the original spiking neural network model of Brunel and Wang[67], using values from neurophysiological data in order to achieve biophysical realism.

In Eqs. (5) and (6), $v_i$ is uncorrelated standard Gaussian noise with an amplitude of $\sigma = 0.01$ (nA). Indeed, the parameters of the DMF model were adjusted such that when isolated (i.e. uncoupled), it describes spontaneous noisy low firing rate activity (3 Hz for the excitatory neurons and 9 Hz for the inhibitory neurons). In our case, the system comprises $N = 90$ cortical and subcortical areas, as detailed above. The feedback inhibition weight, $J_i$, is adjusted for each node $i$ so that the firing rate of the local excitatory neural population is clamped around 3 Hz, whenever nodes are connected or not—this regulation is known as Feedback Inhibition Control (FIC) and the algorithm to achieve this is described in Deco and colleagues[65]. It has been demonstrated that the FIC constrain leads to a better prediction of the resting FC and a more realistic network evoked activity[65]. The excitatory pools are coupled by long-range connections based on the dMRI Structural Matrix $C_{ij}$. The structural matrix $C_{ij}$ denotes the density of fibres between brain areas $i$ and $j$ and is scaled by a global scaling factor G (global conductivity parameter scaling equally all excitatory synapses).

The global scaling factor is the only free parameter of the model. This is adjusted to move the system to its optimal working point, defined by the point where the simulated functional dynamics maximally fits the empirical functional dynamics. More specifically, this global scaling factor is a control parameter that is adjusted to the empirical resting state spatiotemporal dynamics, in order to fit: (1) the grand averaged static FC; (2) the spatiotemporal fluctuations in terms of the metastability (see below). In other words, when we use the standard DMF model, we only perform a global optimization of G and assume consequently that all nodes show the same homogeneous dynamics and that the conductivities of the coupling

of each connecting fibre tract is also globally the same (namely defined by the scaling $G$). Here, we model as the group level but we have shown that the information carried by individual neuroimaging data is robust and reliable[68].

BOLD-fMRI signal: The simulation of the fMRI BOLD-signal in the global brain model is computed by means of the Balloon–Windkessel hemodynamic model[69,70]. The Balloon–Windkessel model describes the coupling of perfusion to BOLD signal, with a dynamical model of the transduction of neural activity into perfusion changes. The model assumes that the BOLD signal is a static nonlinear function of the normalized total deoxyhemoglobin voxel content, normalized venous volume, resting net oxygen extraction fraction by the capillary bed, and resting blood volume fraction. The BOLD-signal estimation for each brain area is computed by the level of neuronal activity summed over all neurons in both populations (excitatory and inhibitory populations) in that particular area. In all our simulation shown here this level of neuronal activity is given by the rate of spiking activity in windows of 1 ms. In brief, for the $i$-th region, neuronal activity $z_i$ causes an increase in a vasodilatory signal $s_i$ that is subject to autoregulatory feedback. Inflow $f_i$ responds in proportion to this signal with concomitant changes in blood volume $v_i$ and deoxyhemoglobin content $q_i$. The equations relating these biophysical variables are:

$$\mathrm{d}s_n/\mathrm{d}t = 0.5r_n^{(E)} + 3 - ks_n - \gamma(f_n - 1) \tag{7}$$

$$\mathrm{d}f_n/\mathrm{d}t = s_n \tag{8}$$

$$\tau\, \mathrm{d}v_n/\mathrm{d}t = f_n - v_n^{\alpha-1} \tag{9}$$

$$\tau\, \mathrm{d}q_n/\mathrm{d}t = f_n(1-\rho)^{f_n^{-1}}/\rho - q_n v_n^{\alpha-1}/v_n \tag{10}$$

where $\rho$ is the resting oxygen extraction fraction. We modified the dependence on the firing rate $z_i$ in Eq. (7) linearly such that the modulation values under task condition are in the experimental range. The BOLD signal in each area $n$, $B_n$, is a static nonlinear function of volume, $v_n$, and deoxyhemoglobin that comprises a volume-weighted sum of extra- and intravascular signals:

$$B_n = V_0[k_1(1-q_n) + k_2(1-q_n/v_n) + k_3(1-v_n)] \tag{11}$$

The biophysical parameters were based on those found in the work of Stephan and colleagues[70]. We chose to concentrate on the frequency range where resting-state activity appears the most functionally relevant, both empirical and simulated BOLD signals were band pass filtered between 0.1 and 0.01 Hz[7,71,72].

Grand average functional connectivity: The grand average FC is defined as the matrix of correlations of the BOLD signals between two brain areas over the whole time window of acquisition. We compare the empirical and simulated FC by the Pearson correlation.

Kuramoto synchronisation index: We measure the synchronisation index as the mean of the Kuramoto order parameter across time. In order to compute the global level of synchronisation, we detrended and extracted the phases of the fMRI time series of each of the 90 brain regions. The Hilbert transform (HT) was applied to the filtered BOLD signals to obtain the associated analytical signals. The analytic signal represents a narrowband signal, $a(t)$, in the time domain as a rotating vector with an instantaneous phase, $\varphi(t)$, and an instantaneous amplitude, $A(t)$, i.e. $a(t) = A(t)\cos(\varphi(t))$. The phase and the amplitude are given by the argument and the modulus, respectively, of the complex signal $z(t)$, given by $z(t) = a(t) + j\cdot \mathrm{HT}[a(t)]$, where $j$ is the imaginary unit. Note that narrowband filtering is a requirement for obtaining meaningful phases and envelopes through the HT.

The global level of phase synchrony was quantified by the Kuramoto order parameter, $R(t)$, given by:

$$R(t) = \left| \sum_{k=1}^{n} e^{i\varphi_k(t)} \right| / n \tag{12}$$

where $n$ is the number of regions in the model (here $n = 90$ for AAL). Thus, $R$ is the average phase of the system and takes the values 0 and 1 for the completely asynchronous and completely synchronized cases, respectively. We compute the difference between the synchronisation index of the model and the empirical data in order to find the optimal working point of the global coupling parameter, $G$, of the model. It has already been shown that the Kuramoto order parameter is excellent for constraining the dynamical working point of whole-brain models fitting empirical neuroimaging data[73]. Similarly, this research also showed that at the optimal dynamic working point of the Kuramoto order parameter, there was also had a good fitting with the FC matrix and more importantly a good fit with the dynamical FC (FCD), which characterizes not only the static spatial correlations (like the FC) but also the spatiotemporal fluctuations. It has also been shown that there is a good correspondence between FC and FC based on Kuramoto. In addition, in the present study, we confirmed that the Kuramoto parameter is consistent with a good model fit to the FC and FCD matrices of the empirical data.

**(B) Extraction of spatiotemporal dynamics structure (spacetime motifs).** For the extraction of the spacetime motifs, we generalised to the whole-brain level established methods for detecting neuronal assemblies from spike data as used by Lopes-dos-Santos and colleagues[20]. Briefly summarising our method in three main steps: (1) construction of event matrix, where events are binned according to different timescales (using a point-process estimation); (2) determination of the number of spacetime motifs, where a null hypothesis distribution based on random matrix theory, the so-called Marčenko–Pastur distribution is generated[22]; and (3) extraction of spacetime motifs using ICA of the Marčenko–Pastur distribution and estimation of corresponding activity, where co-activation patterns are found and used to track the activity over time of the spacetime motifs.

The procedure for extracting spacetime motifs is summarised in Fig. 2. In order to study systematically the relevant timescale, we perform a binning of the data by averaging the neuronal signals in slicing windows of a given width that fix the timescale at that particular level. For this, we utilized the well-established point-process binarisation algorithm for BOLD and MEG signals[18,19,21]. We apply this to the averaged time bin neuronal data from the model (fitted to the BOLD signal) and we also apply this to the beamformed MEG data averaged over all narrow-bands in averaged time bins.

In the procedure, an event for a given brain region is defined by binarising the transformed averaged time bin neuronal time series into z-scores $z_i(t)$ and imposing a threshold $\theta$ such that the binary sequence $\sigma_i(t) = 1$ if $z_i(t) > \theta$, and is crossing the threshold from below, and $\sigma_i(t) = 0$ otherwise. Next, the extracted event matrix (with dimension number of regions × binned time points) is normalized by z-score transformation:

$$e_{ib} = [\sigma_{ib} - \langle \sigma_{ib} \rangle]/\mathrm{std}(\sigma_{ib}) \tag{13}$$

where $e_{ib}$ is the z-scored event count of brain region $i$ in time bin $b$, $\sigma_{ib}$ is the number of events of brain region $i$ in bin $b$, $\langle \sigma_{ib} \rangle$ is the mean event count of brain region $i$ over all time bins, and $\mathrm{std}(\sigma_{ib})$ is the standard deviation of the event counts of brain region $i$ over bins. Thus, in the z-scored event matrix each brain region is set to have null mean and unitary variance.

This procedure has been shown by Tagliazucchi and colleagues[21] to be threshold independent given that the binarisation results from a Poincaré section, which is a classical method for reducing the dimensionality of a dynamical system by analysing the set of points which are the coordinates of the successive intersections of the secant Poincaré plane by the phase space trajectories.

For extracting the spacetime motifs, we applied the method for detecting neuronal assemblies from spike data introduced by Lopes-dos-Santos and colleagues[20] but here we used at the whole-brain level (after the binning of the binarised data). This method first determines the number of neuronal assemblies using eigenvalue analysis for determining the statistical significance of assembly patterns, as introduced by Peyrache and colleagues[75]. The key idea is to estimate the number of assemblies (subsets of brain regions with correlated activity, i.e. what we here call spacetime motifs) by finding the number of principal components of the event matrix with significantly large eigenvalues. For that, we compute the principal components by finding the eigenvectors and corresponding eigenvalues of the covariance matrix of the event matrix $ee^T/N_B$ (where $e$ is the z-scored event matrix with elements $e_{ib}$ and $N_B$ is the number of time bins). The number of spacetime motifs is given by the number of significant components, i.e. significant larger eigenvalues compared to the null hypothesis given by the Marčenko–Pastur distribution. Indeed, Marčenko and Pastur[22] demonstrated that the eigenvalues of the correlation matrix of a normal random matrix M with statistically independent rows follow a probability function described by:

$$p(\lambda) = \frac{q}{2\pi\rho^2} \frac{\sqrt{(\lambda_{\max} - \lambda)(\lambda - \lambda_{\min})}}{\lambda} \tag{14}$$

with $q = N_{\mathrm{columns}}/N_{\mathrm{rows}} \geq 1$, where $\rho^2$ is the variance of the elements of M (in our case $\rho^2 = 1$ due to z-score normalization), $N_{\mathrm{columns}}$ is the number of columns and $N_{\mathrm{rows}}$ the number of rows. $\lambda_{\max}$ and $\lambda_{\min}$ are the maximum and minimum bounds, respectively, and are calculated as:

$$\lambda_{\min}^{\max} = \rho^2 \left(1 \pm \sqrt{1/q}\right)^2 \tag{15}$$

This probability function has finite support given by the interval $\lambda_{\min} \geq \lambda \geq \lambda_{\max}$. Thus, if the rows of M are statistically independent, the probability of finding an eigenvalue outside these bounds is zero. In other words, the variance of the data in any axis cannot be larger than $\lambda_{\max}$ when brain regions are uncorrelated. Therefore, $\lambda_{\max}$ can be used as a statistical threshold for detecting assembly activity[75]. That is, the number of eigenvalues above $\lambda_{\max}$ is used to estimate the number of spacetime motifs (assemblies in the event matrix).

Knowing the number of spacetime motifs, i.e. significant brain assemblies, one can extract them explicitly by using ICA. Here, we employ the FastICA algorithm[74] (as implemented in the FastICA toolbox for MATLAB, http://research.ics.aalto.fi/ica/fastica/). More precisely, we compute the spacetime motifs by applying ICA to the dimensionality-reduced event matrix $e$ obtained by projecting $e$ onto the subspace spanned by the significant principal components (according the procedure explained above). We will denote the significant components, i.e. the

spacetime motifs or brain assemblies, by a matrix $w_{ic}$ where $i$ corresponds to a brain region and $c$ the ICA component.

We can use the spacetime motifs to compute the time course of each assembly activity with single-bin resolution. The activity of each assembly $c$ ($\vec{w}_c$ one of the extracted ICA components of the reduced event matrix, i.e. one column of the matrix $w_{ic}$) can be estimated by projecting the columns of the event matrix onto the axis spanned by that spacetime motif. The projection is defined as the square of the projection length, which can be calculated as:

$$A_{cb} = \mathbf{e}_b^T P_c \mathbf{e}_b \qquad (16)$$

where the projection matrix $P$ is defined as:

$$P_c = \vec{w}_c \otimes \vec{w}_c = \vec{w}_c \vec{w}_c^T \qquad (17)$$

where $\otimes$ is the outer product operator, $\mathbf{e}_b$ is the $b$ column of the event matrix (events at time bin $b$). At each time bin, the length of the projection is a measure of the similarity between the activity of the whole brain regions and the spacetime motif.

Furthermore, we can compute the probability of activation of one particular spacetime motif $c$ by:

$$p(c) = \sum_b A_{cb} / \sum_{c,b} A_{cb} \qquad (18)$$

Entropy of the spatiotemporal dynamical complexity of spacetime motifs: Having computed the spacetime motifs and the associated probability of each pattern at a given timescale allows us to compute the entropy of the occurrence of the spacetime motifs. Given that these spacetime motifs provide a convenient quantitative measure of the dynamical repertoire, the measure of entropy given by the following equation is an excellent way to characterise the richness of the repertoire of spacetime motifs:

$$H = -\sum_c p(c)\log(p(c)) \qquad (19)$$

The entropy characterizes the richness of the switching activity between different spacetime motifs. Complementary to this measure we also introduced a novel, unrelated measure of hierarchical organisation, as a variation on our previous ignition-based hierarchical measures[18,19]. The hierarchy expresses the relevancy of each brain region for the broadcasting of information across the whole brain. We define the hierarchy as the degree of diversity of brain regions according to the level of cohesiveness, i.e. by the standard deviation of the cohesiveness across all brain areas. The cohesiveness of a single brain region, $i$, is given by:

$$\mathrm{Coh}(i) = \sum_c w_{ic} p(c) \left( \sum_j w_{jc} \right) \qquad (20)$$

In other words, the cohesiveness reflects the degree of participation of a single brain region $i$ in spreading information across the whole brain. Note that the above definition is essentially determined by the summation over spacetime motifs of the product of three things: the degree of participation in a spacetime motif $c$, i.e. importance of that node for that assembly ($w_{ic}$); the probability of that spacetime motif, i.e. its relevance across time ($p(c)$); and the 'broadness' of the spacetime motif defined as its size, i.e. how broad is the spreading of the information (correlation) across the whole brain $\left( \sum_j w_{jc} \right)$.

## Data availability

The code and multimodal neuroimaging data from the experiment are available upon request.

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

## Acknowledgements

G.D. is supported by the Spanish Research Project PSI2016-75688-P (AEI/FEDER, EU), by the European Union's Horizon 2020 Research and Innovation Programme under Grant Agreement Nos. 720270 (HBP SGA1) and 785907 (HBP SGA2), and by the Catalan AGAUR Programme 2017 SGR 1545. M.L.K. is supported by the ERC Consolidator Grant: CAREGIVING (No. 615539), and Center for Music in the Brain, funded by the Danish National Research Foundation (DNRF117).

## Author contributions

G.D. and M.L.K. designed the study, developed the method and wrote the manuscript. J.C. analysed MEG data. All the authors edited the manuscript.

## Additional information

**Competing interests:** The authors declare no competing interests.

