## [Peer Review File · Nature Communications]

Reviewers' comments:

Reviewer #1 (Remarks to the Author):

The authors used whole-brain modelling of fMRI signals at the millisecond scale to investigate the output of their model when binning at various timescales. A method based on independent components, previously used to identify and track neuronal assemblies from extracellular recordings of action potentials, was employed to characterize the richness (diversity) of resulting patterns. The authors found an optimum timescale around 200 milliseconds. I am quite sympathetic about this study, which tackles an important question in a clever manner. I have however a number of major and minor concerns, which I detail below.

MAJOR CONCERNS

- 1) Page 3 line 84: "So, in order to start making progress on resolving the question of timescale". The authors seem unaware of previous studies related to theirs, in particular Ikegaya et al. (2004) and Ribeiro, Ribeiro & Copelli (2015), references [1, 2] below.
- 2) Does the "optimal" resolution vary with the value of the threshold used? How does that answer affect the argument in favor of an "optimal" resolution?
- 3) How dependent are the results on the specific parameters of the model? Is there a range of probable parameters, which would lead to a range of optimal processing scales?
- 4) Where do the parameters come from? Has the model been tested with real spike data?
- 5) How can an optimal timescale co-exist with the power law properties of neuronal avalanches [3-5]?
- 6) Recently published results [6] show that the properties of avalanches duration and size change at different levels of cortical activity synchronization. Even assuming the existence of an "optimal timescale" it is possible that this measure is a function of the cortical state. Since cortical activity assumes different levels of synchronization over time [7], perhaps the optimal temporal scale assumes a certain value for states with low synchronization (REM sleep, moments of high attention), whereas in periods of high synchronization (slow wave sleep, drowsiness) the optimal temporal scale assumes other values, mainly due to the higher incidence of silences in unit activity [8].
- 7) It is important to calculate the coefficient of variation (CV) of the events (post-binarization) used in the analysis [9]. If the CV is constant and similar between the subjects, the authors' argument would be strengthened.
- 8) What is the impact of using only 7-minute records in a single behavioral state? Would different time scales be expected for longer records?
- 9) Is the optimal timescale of ~200ms related to theta rhythm or another neural oscillation? What is the real-world neurophysiological significance of the results?
- 10) Why call the patterns Brain Songs? Yuste has claimed the name cortical songs before for a related concept. The patterns detected in the present study seem much more like chords to me.
- 11) Page 8 line 212: "In our case, the system comprises N=90 cortical and subcortical areas, as detailed above". The cortical/subcortical partition was not really detailed, please do so.

MINOR CONCERNS

Page 18 line 537: In order to extract the number of significant cortical assemblies, 'brain songs'. Why cortical only?

Furthermore, there are many typos and misspellings in the text, identified below:

Page 2 line 41: these timescale

Page 7 line 178: networks (E-I networks), which are mutually interconnected according the underlying anatomical

Page 7 line 202: The population firing rates are sigmoid functions ($H(I)$ and $H(E)$) of the input synaptic currents to the excitatory or inhibitory population i is given by $I(I)$ and $I(E)$.

Page 10 line 289 of event matrix, where events are binned according different timescales

Page 11 line 319 The number of brain songs are given by the number of significant components, i.e. significant largely eigenvalues

Page 12 Line 380: information across the whole brain. Note, that the above definition

Page 13 line 394: The data was analysed using a new method, 'brain songs',

Page 14 line 442: timescales of 10ms, 1000ms and 2000ms, the distribution of individual states are much less evenly

Page 14 line 454: As can be seen, the networks are very similar but of course the underlying state probability are different.

Page 14 line 457: The results presented here offer a potential solution to extracting both time and space structure FROM the richness of the repertoire involved in brain processing.

Page 15 Line 477: Yet, as we show here, the relevant timescale is not at the scale of tens of seconds but 200ms which has been found previously using in microstates of EEG

REFERENCES

- 1) Ikegaya Y, Aaron G, Cossart R, Aronov D, Lampl I, Ferster D, Yuste R. Synfire chains and cortical songs: temporal modules of cortical activity. *Science* 304(5670):559-64 (2004)
- 2) Ribeiro T. L., Ribeiro S. & Copelli M. Repertoires of Spike Avalanches Are Modulated by Behavior and Novelty *Front Neural Circuits*. 2016 Mar 22;10:16. doi: 10.3389/fncir.2016.00016. eCollection 2016. (2015)
- 3) Beggs, J. M. & Plenz, D. Neuronal avalanches in neocortical circuits. *J. Neurosci.* 23, 11167–11177 (2003).
- 4) Ribeiro, T. L. et al. Spike avalanches exhibit universal dynamics across the sleep-wake cycle. *PLoS One* 5, e14129 (2010).
- 5) Tagliazucchi, E., Balenzuela, P., Fraiman, D. & Chialvo, D. R. Criticality in large-scale brain fMRI dynamics unveiled by a novel point process analysis. *Front. Physiol.* 3, 15 (2012).
- 6) Hahn, G. et al. Spontaneous cortical activity is transiently poised close to criticality. *PLoS Comput. Biol.* 13, e1005543 (2017).
- 7) Harris, K. D. & Thiele, A. Cortical state and attention. *Nat. Rev. Neurosci.* 12, 509–523 (2011).
- 8) Mochol, G., Hermoso-Mendizabal, A., Sakata, S., Harris, K. D. & de la Rocha, J. Stochastic transitions into silence cause noise correlations in cortical circuits. *Proc. Natl. Acad. Sci. U. S. A.* 112, 3529–3534 (2015).
- 9) Renart, A. et al. The asynchronous state in cortical circuits. *Science* 327, 587–590 (2010).

Reviewer #2 (Remarks to the Author):

This study analyses a computation model of human resting-state fMRI data to investigate the spatiotemporal structure of human brain activity. The authors claim to unravel the “relevant time-scale” of human brain processing. The paper addresses an interesting and timely topic and presents interesting findings. However, overall the study falls short in supporting its strong claims and the conceptual advance of the study is limited by several substantial shortcomings:

- 1) The study provides no mechanistic insights into brain dynamics or even the dynamics of the analyzed computation model. The authors report statistical properties of the analyzed model (e.g., the peak entropy around 200 ms), but no insights are generated on the mechanisms underlying these properties. Why is entropy peaking around 200 ms? Which network features are critical for this?
- 2) The reported results are poorly constrained by empirical data. Ultimately, it remains unclear to what extent the reported findings merely reflect particularities of the analyzed computation model rather than genuine properties of the brain itself. The employed methods should be directly applied to empirical data (e.g., MEG) to substantiate the strong conclusions drawn by the authors.
- 3) The reported analyses are rather superficial. How do the reported results depend on the various parameters and specifics of the applied model and analysis pipeline? How robust are the results?
- 4) The study lacks a link to behavior. A central concept of the study is the ‘relevant timescale’. Relevance for what? Why should the reported effects have any relevance for brain function? I feel that a link to behavior, i.e. to brain function, is required, if one wants to draw conclusions about “relevance”.
- 5) Along the same line, the term ‘relevance’ is poorly defined. The authors start the paper without any definition of this key term. Only later in the introduction (line 63) they start discussing this term, basically stating that ‘relevance’ is a relative term effectively leaving the reader without any definition.
- 6) It seems that at the heart of the analysis is simply a PCA to reduce the dimensionality of the data to its degrees of freedom and then a standard temporal ICA (all on thresholded data). This does not seem to novel to me. I suggest to simply describe what is done in technical terms, rather than inventing new terms, such as e.g. ‘brain songs’.
- 7) The term ‘brain songs’ seems misleading to me, because it implies a spatiotemporal structure, e.g. a sequence of activation across the brain. Instead, in the present study, what is termed ‘brain songs’ are simply the spatial pattern of ICA components, without any intrinsic temporal structure.
- 8) Minor:

I suppose on line 188 it should read ‘DTI’ instead of ‘DSI’

What is n in equation (12)?

Equation (19) should not read $p(c)$ on the left, correct?

Reviewer #3 (Remarks to the Author):

This is an excellent paper by two of the leading scientist in network science. One of the hallmark features of the brain is its spontaneous activity which operates in different time scales. However, different methods like fMRI and EEG/MEG can only depict a certain frequency or time scale range. This makes it rather difficult to conceive the various time scales in conjunction. This is the starting

point of the present paper that introduces a novel method for the analyses of all time scales with fMRI including both fast and slow time scales. They suggest the method of binning from milliseconds to seconds; their method called brain songs allows to extract the spatiotemporal dynamics at each time scale. This is supported by independent measures like entropy and hierarchy that can characterize the dynamic repertoire at each point in time and space. All measures applied target a time scale of 200ms as optimum. Combining computational modelling and empirical fMRI data, this is an excellent paper about relevant time scales at the whole brain levels, brain songs are the authors call it, which raises some questions though.

- The case number of subjects is rather low, $n = 16$. I am wondering whether it might not be useful to use some of the available large scale data sets to confirm the point; this is relevant given the fact that the authors aim establishing the optimum time scale of the brain. For that purpose, one would want a larger number of subjects.

- The TR of the resting state fMRI was rather long with 3030ms – any specific reason for that? Given that the authors are interested in millisecond time scale, I would have expected a shorter TR in at least the subsecond range.

- They calculate the Kuramoto synchronization index for functional connectivity in their model and also the MRI data – did they do some validity check for that? I am aware of at least one fMRI paper doing extensively different analyses of the synchronization/phase-based nature of functional connectivity based on Kuramoto (Huang et al. 2017, Cerebral Cortex). The authors in the present study may want to cite that paper and conduct more control analyses to support their Kuramoto analyses.

- There are some typos and missing terms in the manuscript.

- They adopt the idea of the concept of “brain songs” from the apparently used concept of “cortical songs” from the cellular level. This method of cortical songs is just one method paper, are there any subsequent applications of that?

- They introduce entropy to describe the switching between brain songs. I am not fully clear what exactly that means. They need to specify the exact type of entropy they calculate

- I am a little unclear what exactly means “optimum time scale”...- “optimum” for what???? How do they define and operationalize optimum????????????????

- They cite the Tagliazucchi et al. (20120 Frontiers papers; they suggest to analyse the extremes points of variability changes through a dynamic time series – I am not sure how that relates to their current idea of brain songs

- Important authors’ work in the field are not cited. The fMRI papers by the group around He B. and the group around Northoff as well as MEG/EEG work by the group around Palva/Palva should be cited. They provide important work on the scale-free dynamics as such in the brain’s spontaneous activity and its relevance for behavioral states including self and consciousness. That would nicely complement the excellent discussion in the present paper.

- If different time scale, I was wondering why the authors did not consider scale-free activity measures like PLE and DFA; referring to my previous suggestion.

- They discuss the global workspace theory in the discussion and suggest that the relevant time scale may be 200ms. If they want to link their results to consciousness, they may want to investigate some data set where consciousness is lost (See Tagliazucchi et al. 2016 and Zhang et al. 2018 for a recent paper on scale-freeness and neuronal variability in anesthesia). Moreover competing theories of consciousness like the Integrated Information Theory (IIT) (Tononi et al. 2016) and the Temporo-spatial theory of consciousness (Northoff and Huang 2017, Northoff 2013, 2014) should be discussed, mentioned and cited.

Reviewers' comments:

Reviewer #1 (Remarks to the Author):

The authors used whole-brain modelling of fMRI signals at the millisecond scale to investigate the output of their model when binning at various timescales. A method based on independent components, previously used to identify and track neuronal assemblies from extracellular recordings of action potentials, was employed to characterize the richness (diversity) of resulting patterns. The authors found an optimum timescale around 200 milliseconds. I am quite sympathetic about this study, which tackles an important question in a clever manner.

* Thank you!

I have however a number of major and minor concerns, which I detail below.

MAJOR CONCERNS

1) Page 3 line 84: “So, in order to start making progress on resolving the question of timescale”. The authors seem unaware of previous studies related to theirs, in particular Ikegaya et al. (2004) and Ribeiro, Ribeiro & Copelli (2015), references [1, 2] below.

* We thank the reviewer for these two pertinent references which have now been included and discussed in the revised ms.

2) Does the "optimal" resolution vary with the value of the threshold used? How does that answer affect the argument in favor of an "optimal" resolution?

* Thank you for this important remark that was not fully explained in the ms. In fact, as was previously shown by Tagliazzuchi et al (2012), the binarisation process is fully threshold independent. The reason is because the binarisation emerges as the result of a Poincaré section, and therefore the position of the threshold is irrelevant. We clarify and cite the corresponding literature in the Methods in the revised ms. Furthermore, we ran substantial new simulations for showing this using different thresholds in the new Figure S2 in the supplementary material.

3) *How dependent are the results on the specific parameters of the model? Is there a range of probable parameters, which would lead to a range of optimal processing scales?*

* This is an important question for which we have carried out exhaustive further simulations and added clarifications in the text. In sum, the results depend on two main factors: local node dynamics and whole-brain coupling. In terms of the local node dynamics, we use biophysically realistic parameters (as discussed in the answer to another review question). We have carried out simulations showing the expected (and trivial) result that the timescale depends very directly on these parameters. So biophysical relevant parameters are of course important. More importantly, we have carried out extensive simulations of the whole-brain coupling in terms of shifting the working point of the whole-brain model (Figure S3) and damaging high degree nodes in the whole-brain network (Figure S4). In both cases, the timescale can be significantly affected. In summary, only the realistic whole-brain model consistent with the empirical connectivity data is showing the relevant timescale discovered in this paper.

4) *Where do the parameters come from? Has the model been tested with real spike data?*

* We have expanded on the description of how the parameters are derived using biophysical empirical neurophysiological data and models. In particular, we have clarified that the parameters are from Wong and Wang (2006), which are in turn derived from the original spiking neural network model of Brunel and Wang (2001), using values from neurophysiological data in order to achieve biophysical realism.

5) *How can an optimal timescale co-exist with the power law properties of neuronal avalanches [3-5]?*

* This is a very interesting question. We have added discussion with further references. We suggest that the present timescale findings could be complementary to the important body of work on neuronal avalanches (Plenz, 2007; Ribeiro, 2016; Beggs, 2003; Ribeiro, 2010). Perhaps, as has been suggested, timescale of avalanches is not to be found at the temporal scale of the avalanches, but rather at the scale of their sequences (Ribeiro, 2016). The relevant timescale described here could be related to this scale of sequences of neuronal avalanches. Even more, these two phenomena, criticality of neuronal avalanches and optimal timescale coexist at the subcritical working point of the whole-brain model (Haimovici, 2013; Haimovici, 2016; Deco, 2012).

6) *Recently published results [6] show that the properties of avalanches duration and size change at different levels of cortical activity synchronization. Even assuming the existence of an "optimal timescale" it is possible that this measure is a function of the cortical state. Since cortical activity assumes different levels of synchronization over time [7], perhaps the optimal temporal scale assumes a certain value for states with low synchronization (REM sleep, moments of high attention), whereas in periods of high synchronization (slow wave sleep, drowsiness) the optimal temporal scale assumes other values, mainly due to the higher incidence of silences in unit activity [8].*

* Thank you, we fully agree and are currently working on exactly this question. We have added this important point to the discussion as a further perspective.

7) *It is important to calculate the coefficient of variation (CV) of the events (post-binarization) used in the analysis [9]. If the CV is constant and similar between the subjects, the authors' argument would be strengthened.*

* Reliability at the individual level is an important question. The model used in the ms is based on group data but we have extensively studied the question elsewhere and demonstrated that the information carried by the whole-brain FC matrix is robust and reliable (Pannunzi et al 2017).

8) *What is the impact of using only 7-minute records in a single behavioral state? Would different time scales be expected for longer records?*

* We found the same results having carried out a completely new analysis on 100 unrelated HCP subjects with 15 minutes resting state and a much faster TR of 0.72. The results are shown in Figure S1.

9) *Is the optimal timescale of ~200ms related to theta rhythm or another neural oscillation?*

* This is an important question but difficult to answer with fMRI which obviously does not contain fast oscillations but only around 0.05Hz, which is much slower than theta. The whole-brain dynamic mean field model that we use is asynchronous and the fact that the timescale emerges would argue against theta playing a significant role. We have added these speculations in the revised ms.

9b) *What is the real-world neurophysiological significance of the results?*

Thank you for this interesting question for which we have added *task* data from 100 unrelated HCP subjects. In Figure 6 of the revised ms we show that a very similar timescale emerge, suggesting that the 200 ms timescale must play a key role in awake cognition.

10) *Why call the patterns Brain Songs? Yuste has claimed the name cortical songs before for a related concept. The patterns detected in the present study seem much more like chords to me.*

* We agree with your concern and have renamed the procedure through the ms to use the term “whole-brain spacetime motifs” which more directly captures the technical nature of the methodology. Still, with poetic license we have retained the phrase “brain songs” in the title and abstract, whilst making it clear the relation to the more technical term “whole-brain spacetime motifs” and citing previous conceptualisations including that of Yuste.

11) *Page 8 line 212: “In our case, the system comprises $N=90$ cortical and subcortical areas, as detailed above”. The cortical/subcortical partition was not really detailed, please do so.*

* We have added more information on the widely used AAL parcellation.

MINOR CONCERNS

Page 18 line 537: In order to extract the number of significant cortical assemblies, ‘brain songs’. Why cortical only?

* We have corrected this typo.

Furthermore, there are many typos and misspellings in the text, identified below:

* All have been corrected, thank you!

Page 2 line 41: these timescale

Page 7 line 178: networks (E–I networks), which are mutually interconnected according the underlying anatomical

Page 7 line 202: The population firing rates are sigmoid functions ($H(I)$ and $H(E)$) of the input synaptic currents to the excitatory or inhibitory population i is given by $I(I)$ and $I(E)$.

Page 10 line 289 of event matrix, where events are binned according different timescales

Page 11 line 319 The number of brain songs are given by the number of significant components, i.e. significant largely eigenvalues

Page 12 Line 380: information across the whole brain. Note, that the above definition

Page 13 line 394: The data was analysed using a new method, ‘brain songs’,

Page 14 line 442: timescales of 10ms, 1000ms and 2000ms, the distribution of individual states are much less evenly

Page 14 line 454: As can be seen, the networks are very similar but of course the underlying state probability are different.

Page 14 line 457: *The results presented here offer a potential solution to extracting both time and space structure FROM the richness of the repertoire involved in brain processing.*

Page 15 Line 477: *Yet, as we show here, the relevant timescale is not at the scale of tens of seconds but 200ms which has been found previously using in microstates of EEG*

REFERENCES

- 1) Ikegaya Y, Aaron G, Cossart R, Aronov D, Lampl I, Ferster D, Yuste R. Synfire chains and cortical songs: temporal modules of cortical activity. *Science* 304(5670):559-64 (2004)
- 2) Ribeiro T. L., Ribeiro S. & Copelli M. Repertoires of Spike Avalanches Are Modulated by Behavior and Novelty *Front Neural Circuits*. 2016 Mar 22;10:16. doi: 10.3389/fncir.2016.00016. eCollection 2016. (2015)
- 3) Beggs, J. M. & Plenz, D. Neuronal avalanches in neocortical circuits. *J. Neurosci.* 23, 11167–11177 (2003).
- 4) Ribeiro, T. L. et al. Spike avalanches exhibit universal dynamics across the sleep-wake cycle. *PLoS One* 5, e14129 (2010).
- 5) Tagliazucchi, E., Balenzuela, P., Fraiman, D. & Chialvo, D. R. Criticality in large-scale brain fMRI dynamics unveiled by a novel point process analysis. *Front. Physiol.* 3, 15 (2012).
- 6) Hahn, G. et al. Spontaneous cortical activity is transiently poised close to criticality. *PLoS Comput. Biol.* 13, e1005543 (2017).
- 7) Harris, K. D. & Thiele, A. Cortical state and attention. *Nat. Rev. Neurosci.* 12, 509–523 (2011).
- 8) Mochol, G., Hermoso-Mendizabal, A., Sakata, S., Harris, K. D. & de la Rocha, J. Stochastic transitions into silence cause noise correlations in cortical circuits. *Proc. Natl. Acad. Sci. U. S. A.* 112, 3529–3534 (2015).
- 9) Renart, A. et al. . *Science* 327, 587–590 (2010).

Reviewer #2 (Remarks to the Author):

This study analyses a computation model of human resting-state fMRI data to investigate the spatiotemporal structure of human brain activity. The authors claim to unravel the “relevant time-scale” of human brain processing. The paper addresses an interesting and timely topic and presents interesting findings.

* Thank you!

However, overall the study falls short in supporting its strong claims and the conceptual advance of the study is limited by several substantial shortcomings:

1) The study provides no mechanistic insights into brain dynamics or even the dynamics of the analyzed computation model. The authors report statistical properties of the analyze model (e.g., the peak entropy around 200 ms), but no insights are generated on the mechanisms underlying these properties. Why is entropy peaking around 200 ms? Which network features are critical for this?

* Thank you for this comment which we have addressed with extensive new simulation studies; the results of which have been added to the revised ms and two new supplementary figures. In essence, these simulations have given us further insights into the mechanisms underlying the timescale. We found that shifting the working point of the whole-brain model (shown in Figure S3) will destroy the location in time of the peak of the optimal timescale. Similarly, damaging the edges in the connectivity matrix of the whole-brain network (shown in Figure S4) will also destroy the timescale. This gives us the mechanistic insights that the timescale depends on being in the right dynamical working point and that the anatomical coupling of the human brain is fundamental to having an optimal timescale for information processing. We also carried out additional analyses in task which gave a similar optimal timescale (new Figure 6), suggesting that the timescale is invariant across a brain state and dependent on anatomical structural links rather than functional dynamics.

2) *The reported results are poorly constrained by empirical data. Ultimately, it remains unclear to what extent the reported findings merely reflect particularities of the analyzed computation model rather than genuine properties of the brain itself.*

* In the revised ms, we have clarified that the reported results are highly constrained by the empirical data and have carried out extensive simulation studies as mentioned above. We show that the method works, not only with 'normal' fMRI BOLD data but that they also replicate in state-of-the-art fMRI HCP data in both resting state (new Figure S1) and task (new Figure 6).

2b) *The employed methods should be directly applied to empirical data (e.g., MEG) to substantiate the strong conclusions drawn by the authors.*

* We have followed your excellent advice and spent significant amount of time and effort to apply the second step of our new methodology, namely extracting whole-brain spacetime motifs from MEG data. The excellent time resolution of MEG means that we do not have to use the first step involving constructing a whole-brain model. The results in Figure 7 crucially validate the novel first step of our procedure using whole-brain modelling for accessing much faster timescales using slow fMRI BOLD data. Thus we have confirmed the main timescale finding in a model-free way.

3) *The reported analyses are rather superficial. How do the reported results depend on the various parameters and specifics of the applied model and analysis pipeline? How robust are the results?*

* As stated above, we have carried out many additional simulations testing the robustness of the results (shown in the 6 new figures resuming all the new studies and confirming our results).

4) *The study lacks a link to behavior. A central concept of the study is the 'relevant timescale'. Relevance for what? Why should the reported effects have any relevance for brain function? I feel that a link to behavior, i.e. to brain function, is required, if one wants to draw conclusions about "relevance".*

* We have further expanded on the notion of relevancy and now state that the 'key question investigated here is finding the relevant timescale for obtaining *spacetime motifs*, the spatiotemporal structures underlying whole-brain dynamics'. We also followed your suggestion and have carried out extensive simulation studies in the *task* data from the 100 unrelated HCP subjects. The new Figure 6 shows the entropy and hierarchy measured as a function of the timescale (binning size). The results show a similar peak around 200 ms to that found in resting state data (compare with Figure 3A and Figure S1). This confirms that the timescale found in resting state has direct behavioural relevancy for task.

5) *Along the same line, the term 'relevance' is poorly defined. The authors start the paper without any definition of this key term. Only later in the introduction (line 63) they start discussing this term, basically stating that 'relevance' is a relative term effectively leaving the reader without any definition.*

* We agree, please see our previous answer.

6) *It seems that at the heart of the analysis is simply a PCA to reduce the dimensionality of the data to its degrees of freedom and then a standard temporal ICA (all on thresholded data). This does not seem to novel to me. I suggest to simply describe what is done in technical terms, rather than inventing new terms, such as e.g. 'brain songs'.*

* We have clarified that our methodology is novel in two important ways: 1) using whole-brain computational modelling of neuroimaging timeseries to recover the underlying neurodynamical timeseries of the data (in milliseconds) and 2) estimating the significant whole-brain spacetime motifs emerging at a given timescale and to use the whole-brain measures of entropy and hierarchy to estimate the relevance and richness of the underlying dynamical repertoire. We are not aware of any other existing method able to reconstruct neural timeseries at the millisecond scale from BOLD

timeseries. We are also not aware of any other existing method that can extract spacetime motifs from the whole-brain spatiotemporal patterns at different timescales. As we now show, this extraction method is very useful for analysing e.g. MEG data (see new Figure 7, which confirmed our results). Our method is using state-of-the-art statistical methods to allow a systematic investigation of many timescales from milliseconds to seconds. Furthermore, using this novel methodology, we can study the broadcasting of information across the whole brain over all timescales going beyond existing methods merely estimating static spatial maps, and in a causal way given that we use a causal whole-brain model which can be destroyed.

7) *The term ‘brain songs’ seems misleading to me, because it implies a spatiotemporal structure, e.g. a sequence of activation across the brain. Instead, in the present study, what is termed ‘brain songs’ are simply the spatial pattern of ICA components, without any intrinsic temporal structure.*

* Similar points were also raised by the other two reviewers and we agree that the term ‘brain songs’ is not sufficiently precise and have opted to use the phrase “whole-brain spacetime motifs” which more directly captures the technical nature of the methodology. These motifs are spatiotemporal in nature.

8) *Minor:*

I suppose on line 188 it should read ‘DTI’ instead of ‘DSI’

* We have changed to the more accurate “dMRI”

What is n in equation (12)?

* added

Equation (19) should not read $p(c)$ on the left, correct?

* Thank you for spotting this. Corrected.

Reviewer #3 (Remarks to the Author):

This is an excellent paper by two of the leading scientist in network science.

* Thank you!

One of the hallmark features of the brain is its spontaneous activity which operates in different time scales. However, different methods like fMRI and EEG/MEG can only depict a certain frequency or time scale range. This makes it rather difficult to conceive the various time scales in conjunction. This is the starting point of the present paper that introduces a novel method for the analyses of all time scales with fMRI including both fast and slow time scales. They suggest the method of binning from milliseconds to seconds; their method called brain songs allows to extract the spatiotemporal dynamics at each timescale. This is supported by independent measures like entropy and hierarchy that can characterize the dynamic repertoire at each point in time and space. All measures applied target a time scale of 200ms as optimum. Combining computational modelling and empirical fMRI data, this is an excellent paper about relevant time scales at the whole brain levels, brain songs are the authors call it, which raises some questions though.

* Thank you for your kind comments.

- The case number of subjects is rather low, $n = 16$. I am wondering whether it might not be useful to use some of the available large scale data sets to confirm the point; this is relevant given the fact that the authors aim establishing the optimum time scale of the brain. For that purpose, one would want a larger number of subjects.

* As we now make clear, we deliberately chose a dataset with relatively few participants, long TR and relatively short resting state duration, in order to demonstrate the usefulness of the methodology for the general neuroimaging community. But we also agree that it is important to show in state-of-

the-art data in large number of subjects, so we confirmed the results by repeating the simulations in 100 unrelated HCP subjects for both resting state and task (see new Figures 6 and S1).

- *The TR of the resting state fMRI was rather long with 3030ms – any specific reason for that? Given that the authors are interested in millisecond time scale, I would have expected a shorter TR in at least the subsecond range.*

* Similar to stated in our previous answer, we used a long TR to show the robustness of our methodology but confirmed the results in 100 unrelated HCP subjects with a fast TR of 0.72 seconds.

- *They calculate the Kuramoto synchronization index for functional connectivity in their model and also the MRI data – did they do some validity check for that? I am aware of at least one fMRI paper doing extensively different analyses of the synchronization/phase-based nature of functional connectivity based on Kuramoto (Huang et al. 2017, Cerebral Cortex). The authors in the present study may want to cite that paper and conduct more control analyses to support their Kuramoto analyses.*

* Thank you! Yes, indeed we have previously validated (similar to Huang et al 2017) the consistency between the Kuramoto order parameter and the FC matrix. In Deco et al 2017, we showed that the Kuramoto order parameter is excellent for properly constraining the dynamical working point of whole-brain models. Furthermore, we have shown that it is also consistent with a good fitting of the FC matrix and more importantly with a good fitting of the dynamical FC (FCD) which characterizes not only the static spatial correlations (like the FC) but the spatiotemporal fluctuations. We have now also double checked that the Kuramoto parameter in our present simulations is also consistent with the fitting of the FC and FCD matrices. We clarified this now in the paper and added both references.

- *There are some typos and missing terms in the manuscript.*

* Corrected

- *They adopt the idea of the concept of “brain songs” from the apparently used concept of “cortical songs” from the cellular level. This method of cortical songs is just one method paper, are there any subsequent applications of that?*

* As stated in our reply to the two other reviewers, we have chosen to change the rename the procedure throughout the ms to the term “whole-brain spacetime motifs” which more directly captures the technical nature of the methodology. Still, with poetic license we have retained the phrase “brain songs” only in the title and abstract, whilst making it clear the relation to the more technical term “whole-brain spacetime motifs”.

- *They introduce entropy to describe the switching between brain songs. I am not fully clear what exactly that means. They need to specify the exact type of entropy they calculate ...*

* We describe the entropy more explicitly in the Methods where you can also find the main equation 19 for measuring entropy. In short, the entropy measures the spatiotemporal dynamical complexity of the spacetime motifs.

- *I am a little unclear what exactly means “optimum time scale”...- “optimum” for what???? How do they define and operationalize optimum?????????????*

* Thank you, this is an excellent point. We have further clarified the meaning of relevancy (optimum) should be seen in the context of finding the relevant timescale for obtaining spacetime motifs, the spatiotemporal structures underlying whole-brain dynamics.

- They cite the Tagliazucchi et al. (2012) *Frontiers papers*; they suggest to analyse the extremes points of variability changes through a dynamic time series – I am not sure how that relates to their current idea of brain songs

* We used the point process for binarising the timeseries, which is fully threshold independent. The reason is because the binarisation emerges as the result of a Poincaré section, and therefore the position of the threshold is irrelevant. This step in the processing is summarised in Figure 2A and is required for extraction of the spacetime motifs. This has been clarified in the revised ms.

- Important authors' work in the field are not cited. The fMRI papers by the group around He B. and the group around Northoff as well as MEG/EEG work by the group around Palva/Palva should be cited. They provide important work on the scale-free dynamics as such in the brain's spontaneous activity and its relevance for behavioral states including self and consciousness. That would nicely complement the excellent discussion in the present paper.

* Excellent point! We have added further discussion and references on scale-free dynamics and avalanches in the discussion.

- If different time scale, I was wondering why the authors did not consider scale-free activity measures like PLE and DFA; referring to my previous suggestion.

* We are not using scale-free measures since the existing measures for fitting the whole-brain model to the empirical data does not need this excellent technique. Nevertheless, we of course use conventional detrending methods as specified now in the Methods.

- They discuss the global workspace theory in the discussion and suggest that the relevant time scale may be 200ms. If they want to link their results to consciousness, they may want to investigate some data set where consciousness is lost (See Tagliazucchi et al. 2016 and Zhang et al. 2018 for a recent paper on scale-freeness and neuronal variability in anesthesia). Moreover competing theories of consciousness like the Integrated Information Theory (IIT) (Tononi et al. 2016) and the Temporo-spatial theory of consciousness (Northoff and Huang 2017, Northoff 2013, 2014) should be discussed, mentioned and cited.

* Thank you, we have discussed the findings in the context of these other excellent theories of consciousness.

Reviewers' comments:

Reviewer #1 (Remarks to the Author):

I am fully satisfied with the response of the authors, and with the revised manuscript.

Reviewer #2 (Remarks to the Author):

The revised manuscript is significantly improved and addresses several issues raised in the initial review. Most importantly, the revised manuscript includes a new analysis of MEG data that interestingly yields very similar results as the analysis of simulated brain activity. I feel that in particular these new results add to the paper. Furthermore, the revised manuscript includes two control analysis that show that the key results are lost if the simulated network is severely altered.

Despite these improvements, I feel that there are still several significant shortcomings:

1) I still feel that the paper stays rather descriptive and that the mechanistic insight provided by the results is limited. Why does the entropy of the independent components peak around 200ms for the model or the MEG data? The fact that this peak is lost if the model is severely altered (Fig. S3, S4) shows that this feature is not robust to all gross changes of the model, but this is expected and does not provide any real insight into why entropy peaks at this timescale. The authors mention that this peak "trivially" depends on some network parameters that were chosen to be biologically plausible (page 18, para 4), but this is neither further explained nor is any data shown. The same applies to the MEG results. E.g., can the authors rule out that the complexity simply drops towards longer timescales (>200ms) because larger binning hampers the ability to temporally resolve the underlying dynamics and that the drop for shortest timescales (<200ms) results from averaging out high-frequency noise?

2) Along the same line, I previously mentioned that it does not become clear to what extent the present results depend on specific parameter choices and analysis steps. I feel that the revised manuscript is still weak at this point. E.g., the authors mention that they "carried out extensive simulations" to show how the network behavior depends on a particular parameter (page 18, para 4), but only very limited results are shown (Fig. S3) and reported.

3) The authors conclusions about behavioral relevance is not supported by the reported evidence. The authors now report the same finding (peak entropy around 200ms) for simulations based on social cognition fMRI-task data as before for resting state data. However, this finding is not sufficient to support the authors claim that "This confirms that timescale found in resting state has behavioural relevancy for task". In contrast, if any conclusion about behavioral relevance should be based on this very limited evidence, then finding the same statistical property for simulations based on fMRI data from two different behavioral context rather suggests that this property is independent from, and thus potential not relevant for, behavior.

If feel that the manuscript remains conceptually unclear and even misleading at critical points:

4) The authors now use the term 'spacetime motif' instead of 'brain song' (although 'brain song' is also still used at several points, e.g. in the abstract). Again, I feel that the term 'spatiotemporal motif' implies not only a spatial, but also temporal structure, such as e.g. a spatiotemporal sequence of brain activation that is repeatedly expressed in the brain. I feel this is misleading, because what is termed 'spacetime motifs' or 'brain songs' here, is simply independent components of a temporal ICA applied to binned data. These components have a fixed spatial pattern and some temporal activation, but no temporal pattern or structure. I feel that, instead of introducing new terms and jargon, the paper may benefit from simply describing what is computed

in well-known technical terms.

5) This also applies to the description of what is actually studied. E.g., in the introduction it is stated “the key question investigated here is finding the relevant timescale for obtaining spacetime motifs, the spatiotemporal structures underlying whole-brain dynamics. In other words, we are studying the relevant timescale for maximising the richness of repertoire of spacetime motifs.” However, it is neither explained or clearly defined, what is really meant with such vague terms like ‘spacetime motifs’ or ‘richness of repertoire’.

6) Also the use of the term ‘information’ seems misleading. E.g., in the introduction it is stated “Using this novel methodology, we can study broadcasting of information across the whole brain over all timescales going beyond existing methods merely estimating static spatial maps”. The same holds for the first sentence of the discussion “In this paper we have investigated a central question in neuroscience, namely which is the most relevant timescale for brain processing, i.e. broadcasting and making information available across the whole-brain.” The applied analysis all merely deal with statistical properties of simulated brain activity or raw MEG activity unrelated to any other experimental variable, such as sensory inputs, behavior or cognitive variables. Thus, these analyses do not allow for drawing any inferences about information being encoded or broadcasted by the brain. In fact, the described dynamics and statistical properties may be completely unrelated to the brain’s information processing.

7) The applied ‘hierarchy’ measure is unclear. This is not a standard measure. Yet, there is no formal description of what is computed, and the corresponding text is unclear. E.g., it seems as if ‘hierarchy’ is computed for each ‘motif’, but it remains unclear how the results are combined across motifs. Furthermore, and even more importantly, it does not become clear if the applied ‘entropy’ and ‘hierarchy’ measures are really statistically independent. This is implied at several points in the manuscript (e.g., in the abstract), but not really explained or shown. Intuitively, I feel that both measures are in fact not independent, because both measures grow with a more uniform distribution of component (motif) probabilities and higher number of components. Furthermore, why the term ‘hierarchy’? It seems as if the measure quantifies variability, but not hierarchical structure.

8) The paper lacks any inferential statistics. There is not a single p-value or null-hypothesis. E.g., it is stated that the patterns in Figure 4 resemble known resting state networks. However, no test and corresponding p-value is provided for the null-hypothesis that the patterns in Figure 4 are just some (mostly local) patterns.

9) Throughout the manuscript the applied analyses are labeled as a ‘novel method’ or ‘novel methodology’. I agree that the specific sequence of analysis steps has not been applied before, but all the applied steps and methods have been described and used before. I do not see a new method or measure. Thus, I suggest to tone this down and to simply and precisely describe what is done in known technical terms.

10) Why was the MEG data narrow-band filtered between 2 and 40 Hz? How does this filtering affect the results?

11) The discussion of the present findings in the context of criticality and avalanches seems unclear. A key feature of avalanches and/or critical systems is that they are scale-free. This seems to stand in strong contrast to the authors interpretation of the present results (one ‘relevant’ time-scale). It does not become sufficiently clear how this apparent contrast can be reconciled.

12) Page 4, para 1. Without in-depth knowledge of the specific literature, this paragraph is hard to understand.

Reviewer #3 (Remarks to the Author):

My comments have been well addressed. The manuscript is now in an excellent and can be published from my point of view. Well done!

Reviewers' comments:

Reviewer #1 (Remarks to the Author):

I am fully satisfied with the response of the authors, and with the revised manuscript. Thank you very much for your constructive comments.

Reviewer #2 (Remarks to the Author):

The revised manuscript is significantly improved and addresses several issues raised in the initial review. Most importantly, the revised manuscript includes a new analysis of MEG data that interestingly yields very similar results as the analysis of simulated brain activity. I feel that in particular these new results add to the paper. Furthermore, the revised manuscript includes two control analysis that show that the key results are lost if the simulated network is severely altered. Thank you.

Despite these improvements, I feel that there are still several significant shortcomings:

1) I still feel that the paper stays rather descriptive and that the mechanistic insight provided by the results is limited. Why does the entropy of the independent components peak around 200ms for the model or the MEG data? The fact that this peak is lost if the model is severely altered (Fig. S3, S4) shows that this feature is not robust to all gross changes of the model, but this is expected and does not provide any real insight into why entropy peaks at this timescale. The authors mention that this peak “trivially” depends on some network parameters that were chosen to be biologically plausible (page 18, para 4), but this is neither further explained nor is any data shown. The same applies to the MEG results. E.g., can the authors rule out that the complexity simply drops towards longer timescales (>200ms) because larger binning hampers the ability to temporally resolve the underlying dynamics and that the drop for shortest timescales (<200ms) results from averaging out high-frequency noise?

Our timescale findings are based on using a causal mechanistic whole-brain model using differential equations. This means that our findings are not descriptive. When the results change after directly manipulating elements of the mechanistic model, this directly establishes a causal relationship. We manipulate every part of the model to establish a causal relationship between a given element and the findings. Essentially, as shown in the methods (see equations 1-6) there are three main elements to the whole-brain model: 1) the underlying anatomical connectivity, 2) the global working point of the dynamics (one parameter, G) and 3) the local regional dynamics. Following your constructive comments in the first round of reviews, we showed that changing the first two elements (global working point in Figure S4 and connectivity in Figure S5) has a direct causal influence on the timescale where the position of the maximum moves far away from 200ms. Here, we further demonstrate the causal influence of the third element by two manipulations: A) change the biophysical latencies of the NMDA of the local dynamics to a non-biological value and B) change the ratio of excitation to inhibition at the local regional level. This is now shown in the new Supplementary Figure S3. Also, following your comments, we have now clarified how mechanistic insights can be gained by changing causative elements in our whole-brain model.

Furthermore, to investigate any potential effects of binning on the 200ms timescale peak independent of the whole-brain model, we have also carried out further analyses on the MEG data by splitting the data into different (δ , θ , α and β) bands. We show that the MEG data shows a similar timescale peak of around 200ms for all bands, which argues against any potential

strange resolution binning effects. This new figure replaces Figure 7. We have further clarified this important and non-trivial finding in the main ms.

2) Along the same line, I previously mentioned that it does not become clear to what extent the present results depend on specific parameter choices and analysis steps. I feel that the revised manuscript is still weak at this point. E.g., the authors mention that they “carried out extensive simulations” to show how the network behavior depends on a particular parameter (page 18, para 4), but only very limited results are shown (Fig. S3) and reported.

As outlined above, the model essentially has three elements, which we have manipulated and shown their causal effects on the timescale. As we make clear in the Methods, the connectivity is empirically determined, the local dynamics are biophysical realistic and the global working point parameter is parametrically varied to find the optimal working point by fitting the empirical dynamics of the functional connectivity of the fMRI data. This global coupling parameter, G , is the *only* free parameter in the model. Nevertheless we manipulated all three elements to show that each of them has a causal effect on our timescale finding. Furthermore, we also manipulated the threshold of the method, i.e. the Poincaré cut. In addition, in the previous round of revisions we showed that the model is robust when fitting the model to other datasets such as the HCP resting state and task data for 100 participants. We also computed the timescale of MEG data (which additional preprocessing). All of this takes weeks of computations on the cluster - which is what we mean by “extensive simulations”. We have stressed all of these important points in the revised ms.

3) The authors conclusions about behavioral relevance is not supported by the reported evidence. The authors now report the same finding (peak entropy around 200ms) for simulations based on social cognition fMRI-task data as before for resting state data. However, this finding is not sufficient to support the authors claim that “This confirms that timescale found in resting state has behavioural relevancy for task”. In contrast, if any conclusion about behavioral relevance should be based on this very limited evidence, then finding the same statistical property for simulations based on fMRI data from two different behavioral context rather suggests that this property is independent from, and thus potential not relevant for, behavior.

Following the comment regarding behavioural relevance that you raised in the previous version, we analysed the HCP task data for 100 unrelated participants and found a similar timescale peaking at 200ms. In our view, this result speaks to the generality of our timescale finding being conserved across experimental conditions and therefore could be a general principle of brain function. However, it is of course possible that it is not directly relevant to behaviour but just an intrinsic property of brain dynamics, which of course in turn must impose some constraints on behaviour. Still, we agree that our findings do not speak directly to behavioural relevance and we have therefore modified our speculations about behaviour in the revised ms and instead stress the intrinsic character of the timescale for the whole-brain dynamics. In addition, we have added the caveat that the exact relationship between timescale and brain function remains to be clarified, and could be further investigated in altered brain states such as sleep or anaesthesia. We have been in correspondence with the editor, who agrees that a demonstration of behavioural relevance is not needed for this ms.

If feel that the manuscript remains conceptually unclear and even misleading at critical points:

For each of your comments, we have tried to address your concerns to make the ms clearer.

4) The authors now use the term ‘spacetime motif’ instead of ‘brain song’ (although ‘brain song’ is also still used at several points, e.g. in the abstract). Again, I feel that the term ‘spatiotemporal motif’ implies not only a spatial, but also temporal structure, such as e.g. a spatiotemporal sequence of brain activation that is repeatedly expressed in the brain. I feel this is misleading, because what is termed ‘spacetime motifs’ or ‘brain songs’ here, is simply independent components of a temporal ICA applied to binned data. These components have a fixed spatial pattern and some

temporal activation, but no temporal pattern or structure. I feel that, instead of introducing new terms and jargon, the paper may benefit from simply describing what is computed in well-known technical terms.

We followed the constructive comments from all three reviewers in round 1, and have already changed the terminology. The two other reviewers agree that the classical term “spacetime motifs” describes exactly our method, which is describing the temporal evolution of spatial ICA patterns. We use “spacetime motifs” as a shorthand for this process, which is not jargon but described precisely and in full details in the Methods.

5) This also applies to the description of what is actually studied. E.g., in the introduction it is stated “the key question investigated here is finding the relevant timescale for obtaining spacetime motifs, the spatiotemporal structures underlying whole-brain dynamics. In other words, we are studying the relevant timescale for maximising the richness of repertoire of spacetime motifs.” However, it is neither explained or clearly defined, what is really meant with such vague terms like ‘spacetime motifs’ or ‘richness of repertoire’.

We clarify now how the richness of repertoire is referring directly to the equations provided in the Methods.

6) Also the use of the term ‘information’ seems misleading. E.g., in the introduction it is stated “Using this novel methodology, we can study broadcasting of information across the whole brain over all timescales going beyond existing methods merely estimating static spatial maps“. The same holds for the first sentence of the discussion “In this paper we have investigated a central question in neuroscience, namely which is the most relevant timescale for brain processing, i.e. broadcasting and making information available across the whole-brain.” The applied analysis all merely deal with statistical properties of simulated brain activity or raw MEG activity unrelated to any other experimental variable, such as sensory inputs, behavior or cognitive variables. Thus, these analyses do not allow for drawing any inferences about information being encoded or broadcasted by the brain. In fact, the described dynamics and statistical properties may be completely unrelated to the brain’s information processing.

In the revised ms, we clarify our semantic use of the term ‘information’.

7) The applied ‘hierarchy’ measure is unclear. This is not a standard measure. Yet, there is no formal description of what is computed, and the corresponding text is unclear. E.g., it seems as if ‘hierarchy’ is computed for each ‘motif’, but it remains unclear how the results are combined across motifs. Furthermore, and even more importantly, it does not become clear if the applied ‘entropy’ and ‘hierarchy’ measures are really statistically independent. This is implied at several points in the manuscript (e.g., in the abstract), but not really explained or shown. Intuitively, I feel that both measures are in fact not independent, because both measures grow with a more uniform distribution of component (motif) probabilities and higher number of components. Furthermore, why the term ‘hierarchy’? It seems as if the measure quantifies variability, but not hierarchical structure.

We further clarify our ‘hierarchy’ measure which is explicitly described by Equation 20. We also refer the reader to our previous Neuron paper on the topic of hierarchy, which we describe in full details.

8) The paper lacks any inferential statistics. There is not a single p-value or null-hypothesis. E.g., it is stated that the patterns in Figure 4 resemble known resting state networks. However, no test and corresponding p-value is provided for the null-hypothesis that the patterns in Figure 4 are just some (mostly local) patterns.

We are not claiming that the potential relationship to resting state patterns is central to the findings and we are not really expecting them to be the same. We merely show the renderings of the

spacetime motifs such that the reader can see these rendered on the human brain and gauge any potential similarities. We stress this in the revised ms.

9) *Throughout the manuscript the applied analyses are labeled as a ‘novel method’ or ‘novel methodology’. I agree that the specific sequence of analysis steps has not been applied before, but all the applied steps and methods have been described and used before. I do not see a new method or measure. Thus, I suggest to tone this down and to simply and precisely describe what is done in known technical terms.*

The main novelty of the ms is the finding of whole-brain timescale of around 200ms. We have modified the ms to remove ‘novel method’ and instead stress that this result comes about through a novel application of combining existing methods.

10) *Why was the MEG data narrow-band filtered between 2 and 40 Hz? How does this filtering affect the results?*

As mentioned in point 1, we have now carried out further analysis at different classical (delta, theta, alpha and beta) bands of the MEG data. The same timescale finding comes out in each band.

11) *The discussion of the present findings in the context of criticality and avalanches seems unclear. A key feature of avalanches and/or critical systems is that they are scale-free. This seems to stand in strong contrast to the authors interpretation of the present results (one ‘relevant’ time-scale). It does not become sufficiently clear how this apparent contrast can be reconciled.*

We have included this discussion following the comments raised by the other reviewers. The study was not designed to resolve this interesting issue, which is why we are merely discussing potential interpretations. We have made this clearer.

12) *Page 4, para 1. Without in-depth knowledge of the specific literature, this paragraph is hard to understand.*

We have simplified the paragraph, which was written in reply to the comments raised by the other reviewers.

Reviewer #3 (Remarks to the Author):

My comments have been well addressed. The manuscript is now in an excellent and can be published from my point of view. Well done!

Thank you for your excellent comments

REVIEWERS' COMMENTS:

Reviewer #2 (Remarks to the Author):

The authors added further analyses and critically revised the manuscript. This has significantly improved the manuscript and addressed most of my previous concerns.

Nevertheless, I still feel that the following points should be considered to avoid misunderstandings:

I still feel that term 'spacetime motif' (or 'brain songs') is misleading, because it implies structure in space and time. However, what is reported are spatial patterns or networks, without any intrinsic temporal structure. Thus, I feel that terms like 'networks' or 'cortical patterns' are more appropriate. I suggest using these terms.

I still feel that the discussion of the present results in the context of information transfer is misleading. The first sentence of the discussion reads "In this paper we have investigated a central question in neuroscience, namely which is the most relevant timescale for brain processing, i.e. broadcasting and making information available across the whole-brain." Or in the introduction "Using this novel application for extracting spacetime motifs at the whole-brain level, we can study broadcasting of information across the whole brain". There are similar statements in the description of the hierarchy measure. I feel that that these statements are misleading. The present study investigates resting-state patterns of neuronal activity (or simulations thereof) without any relationship to sensory inputs, behavior or cognition. Thus, the reported effects are not related to neural encoding of information about any of these variables. This is in contrast to a large amount of neuroscience research, which indeed investigates exactly this question, i.e. the mechanisms of neural encoding and processing of various types of information. These statements should be removed or revised in order not to mislead the readers.

Minor: Is a word missing after "human" on line 708?